# Recent Advances in Magnetostrictive Tb-Dy-Fe Alloys

**Zijing Yang, Jiheng Li \*** , **Zhiguang Zhou** , **Jiaxin Gong, Xiaoqian Bao and Xuexu Gao**

State Key Laboratory for Advanced Metals and Materials, University of Science and Technology Beijing, Beijing 100083, China; g20209193@xs.ustb.edu.cn (Z.Y.); zzgsunrui@126.com (Z.Z.); gongjiaxin416@163.com (J.G.); bxq118@ustb.edu.cn (X.B.); gaox@skl.ustb.edu.cn (X.G.)

\* Correspondence: lijh@ustb.edu.cn; Tel.: +86-10-6233-3431

**Abstract:** As giant magnetostrictive materials with low magnetocrystalline anisotropy, Tb-Dy-Fe alloys are widely used in transducers, actuators and sensors due to the effective conversion between magnetic energy and mechanical energy (or acoustic energy). However, the intrinsic brittleness of intermetallic compounds leads to their poor machinability and makes them prone to fracture, which limits their practical applications. Recently, the addition of a fourth element to Tb-Dy-Fe alloys, such as Ho, Pr, Co, Nb, Cu and Ti, has been studied to improve their magnetostrictive and mechanical properties. This review starts with a brief introduction to the characteristics of Tb-Dy-Fe alloys and then focuses on the research progress in recent years. First, studies on the crystal growth mechanism in directional solidification, process improvement by introducing a strong magnetic field and the effects of substitute elements are described. Then, meaningful progress in mechanical properties, composite materials, the structural origin of magnetostriction based on ferromagnetic MPB theory and sensor applications are summarized. Furthermore, sintered composite materials based on the reconstruction of the grain boundary phase also provide new ideas for the development of magnetostrictive materials with excellent comprehensive properties, including high magnetostriction, high mechanical properties, high corrosion resistance and high resistivity. Finally, future prospects are presented. This review will be helpful for the design of novel magnetostrictive Tb-Dy-Fe alloys, the improvement of magnetostrictive and mechanical properties and the understanding of magnetostriction mechanisms.

**Keywords:** magnetostriction; Tb-Dy-Fe alloys; directional solidification; mechanical property; Tb-Dy-Fe composites; applications





## 1. Introduction

The physical effect of the magnetostriction of Tb-Dy-Fe alloys is utilized to realize their application in sensors, transducers and actuators through the conversion of magnetoelastic properties and mechanical energy.

Clark et al. [1,2] discovered that the magnetization and magnetocrystalline anisotropy of composite rare-earth compounds composed of $R'Fe_2$ and $R''Fe_2$ ($R'$ and $R''$ denote different rare-earth elements) had a superposition effect. In particular, the $\lambda_{111}$ of pseudobinary $Tb_xDy_{1-x}Fe_2$ compounds ($0 < x < 1$) could reach $1600-2400 \times 10^{-6}$, and the external magnetic field intensity required to achieve saturation magnetization was only $1.6 \times 10^3$ kA/m. Due to the large anisotropy of magnetostriction in the Tb-Dy-Fe single crystal, the magnetostrictive strain in the <111> easy axis is the largest. However, <111> is not the easy growth direction of the crystal. It is necessary to develop directional solidification technology to bring the grain orientation closer to the easy magnetization direction <111> [3]. Tb-Dy-Fe alloys in <110> and <112> orientations are usually prepared by directional solidification [4,5].

During the last years, many efforts have been dedicated to enhancing magnetostriction to reduce costs, such as alloying with other elements and improving the preparation process [6–9]. In previous research, the partial substitution of Tb and Dy by Ho was

investigated to reduce the magnetocrystalline anisotropy and effectively decrease the hysteresis [10]. Some multicomponent alloys, such as $(Tb_{0.7}Dy_{0.3})_{0.7}Pr_{0.3}(Fe_{1-x}Co_x)_{1.85}$ $(0 \leq x \leq 0.6)$ and $Tb_{0.3}Dy_{0.7}(Fe_{1-x}Si_x)_{1.95}$ (x = 0.025), also presented good low-field magnetostriction performance [10–12]. However, intrinsic brittleness and large eddy-current loss at high frequency still limit the application range of Tb-Dy-Fe alloys.

The magnetostriction of Tb-Dy-Fe alloys is related to the magnetocrystalline anisotropy of rare-earth compounds. It is also considered to be derived from the interaction between 4f electrons of rare-earth elements and 3d electrons of transition-metal ions [13]. Since the (Tb, Dy)Fe$_2$ pseudobinary alloy system was proposed, there has been little effective progress in the understanding of its magnetostriction mechanism. To explore the great enhancement of its properties, an in-depth understanding of its physical nature is urgently needed. In recent years, the concept and implication of the morphotropic phase boundary (MPB) have been introduced to the ferromagnetic material system, which provides a new perspective for the research of the magnetostrictive effect of Tb-Dy-Fe alloys and the development of high-performance magnetostrictive materials [14–16]. Furthermore, the emergence and development of a new generation of synchrotron and light sources could more accurately detect the position change of atoms in the crystal, which would be conducive to the study of the magnetostriction mechanism of Tb-Dy-Fe alloys [17,18].

Based on previous works, this review focuses on the research progress of Tb-Dy-Fe alloys in recent years. In Section 2, the study of the crystal growth mechanism in directional solidification and the introduction of a strong magnetic field for process improvement are introduced. The effects of substitute elements are discussed in Section 3, and then the meaningful progress made in recent years in understanding the mechanical properties, composites and ferromagnetic MPB of Tb-Dy-Fe alloys is summarized in Sections 4–6, respectively. Finally, the latest progress in the application of high-sensitivity sensors designed with Tb-Dy-Fe alloys is discussed.

## 2. Grain Orientation and Properties of Directionally Solidified Tb-Dy-Fe Alloys

In order to achieve the large magnetostriction of Tb-Dy-Fe compounds in a low magnetic field, directional solidification technology needs to be used to orient the grains in the easy magnetization direction as much as possible due to the anisotropy of magnetostriction.

### 2.1. Grain Growth and Orientation Control during Directional Solidification Process

The growth process of crystals in directional solidification directly affects the final orientation of grains. Therefore, it is necessary to understand the crystal growth mechanism and orientation selection mechanism during this process; for example, the solid–liquid interface morphology and atomic adhesion kinetics should be researched.

The solid–liquid interface morphology plays a key role in single-crystal growth. By controlling the zone-melting length using a modified optical zone-melting method, Kang et al. [19] obtained three forms of solid–liquid interface morphologies, namely, convex, flat and concave interfaces, as shown in Figure 1. They successfully prepared an <110> axial oriented Tb-Dy-Fe twinned single crystal without radial composition segregation. Although the convex interface was conducive to single-crystal growth, it would produce radial component segregation; that is, the shape of the solid–liquid interface could affect the radial component distribution. By establishing their theoretical models, the effects of the temperature gradient, growth rate and zone-melting length on radial component segregation were qualitatively described.

Generally, solidification parameters, such as the temperature gradient and solidification rate, influence the evolution of texture during the directional solidification process. By comparing the texture at different distances from the onset of solidification, Palit et al. [20] found that the transition in the preferred growth direction from <110> to <112> occurred through intermediate <123> texture components. Furthermore, plane-front solidification morphology and irregular peritectic coupled growth were observed in a wide range of solidification rates (5–80 cm/h), while the preferred direction changed from <311> to

<110> to <112> at 5–100 cm/h solidification rates [21]. The {111} planes in Figure 2a were composed of two different types of atomic layers, in which one layer was all Fe, which increased the obstacle of atomic arrangement in the growth process. At the same time, the {311} planes could be attached to the two {111} planes, as shown in Figure 2b. Therefore, although {111} planes had higher atomic bulk density, the preferred orientation at low solidification rates (5–30 cm/h) was <311>. In addition, the highest magnetostriction was achieved in the sample with a solidification rate of 100 cm/h due to the <112> preferred orientation, as shown in Figure 2c,d.

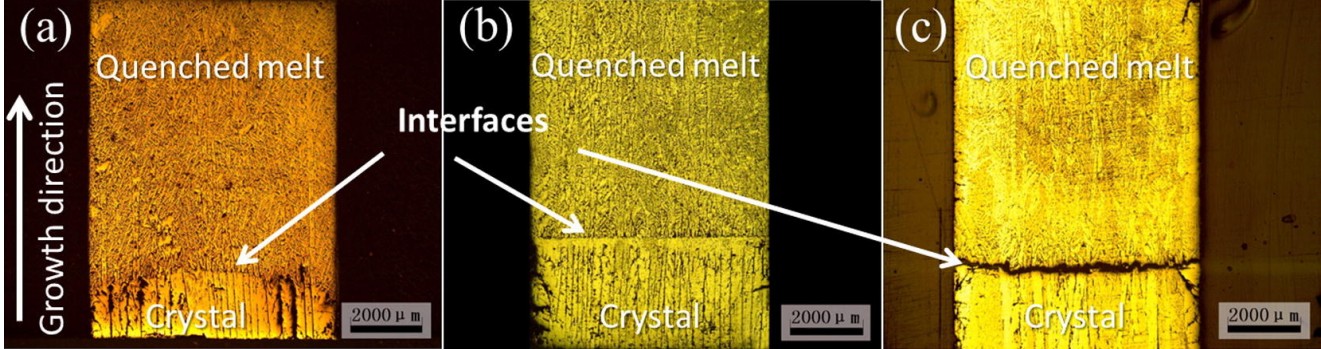

**Figure 1.** Three forms of solid–liquid interface morphologies under the condition of V = 15 mm/h with different half zone-melting lengths: (**a**) P = 60; $L_{1/2}$ = 7 mm (convex); (**b**) P = 70; $L_{1/2}$ = 10 mm (flat); (**c**) P = 80; $L_{1/2}$ = 15 mm (concave) [19] (here, P is heating power) (Reprinted with permission from Ref. [19]. Copyright 2015 Elsevier).

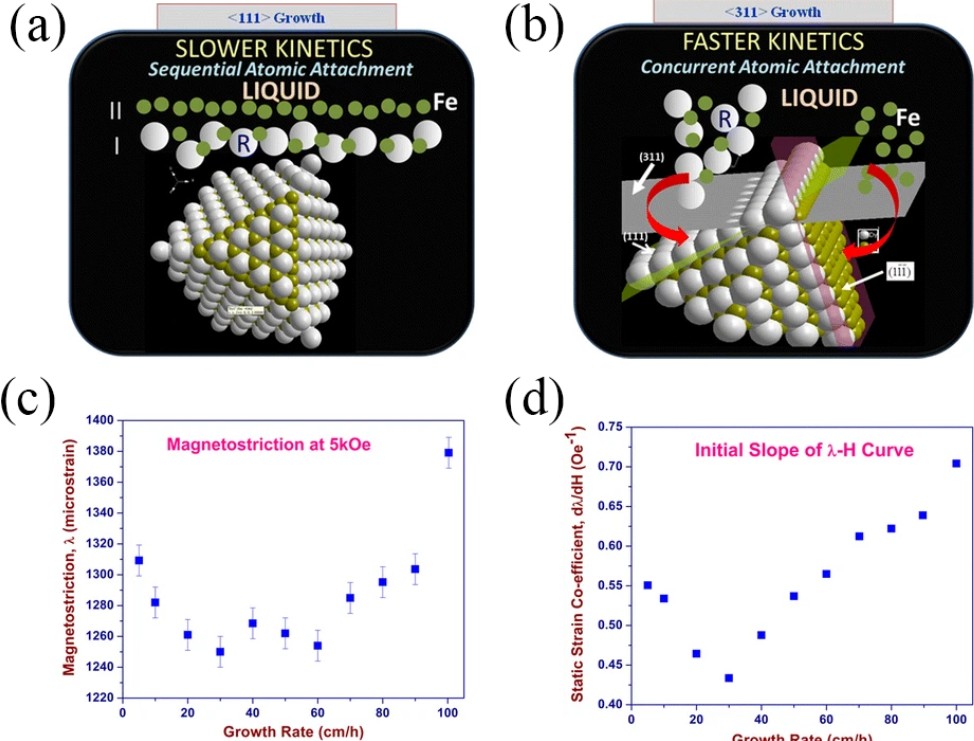

**Figure 2.** Comparison of atomic attachment kinetics at (**a**) (111) and (**b**) (311) interfaces; (**c**) magnetostriction (λ) of directionally solidified samples measured at an applied field of 5 kOe and (**d**) the plot of slope (dλ/dH) of the initial λ-H plot, plotted as a function of growth rate [21] (Adapted with permission from Ref. [21]. 2016 Springer Nature).

The growth twins of Tb-Dy-Fe alloys changed the crystal orientation, which was related to the crystal orientation of mirror symmetry [22]. Previous studies have shown

that different solidification rates correspond to different solidification morphologies; that is, with the increase in the solidification rate, the preferred axial orientation changed from <101> to <112> and was then reoriented to <101> [23]. Recently, 3D spatial extension and transformation of the whole grain were asserted to be the key to interpreting the transformation of the preferred axial orientation. As a polyhedral material with a face-centered cubic structure, Tb-Dy-Fe alloy dendrites grow in the form of twin-related lamellae. In this case, when the crystal grows in a cellular form, the initial dendritic arms of the <110> axially oriented crystal have two extension directions, which will occupy more space and obtain preferential growth (Figure 3). Therefore, the transformation of the preferred axial orientation was explained by the different space-occupying capacities caused by the different morphological configurations for <101> and <112> axially oriented grains [24].

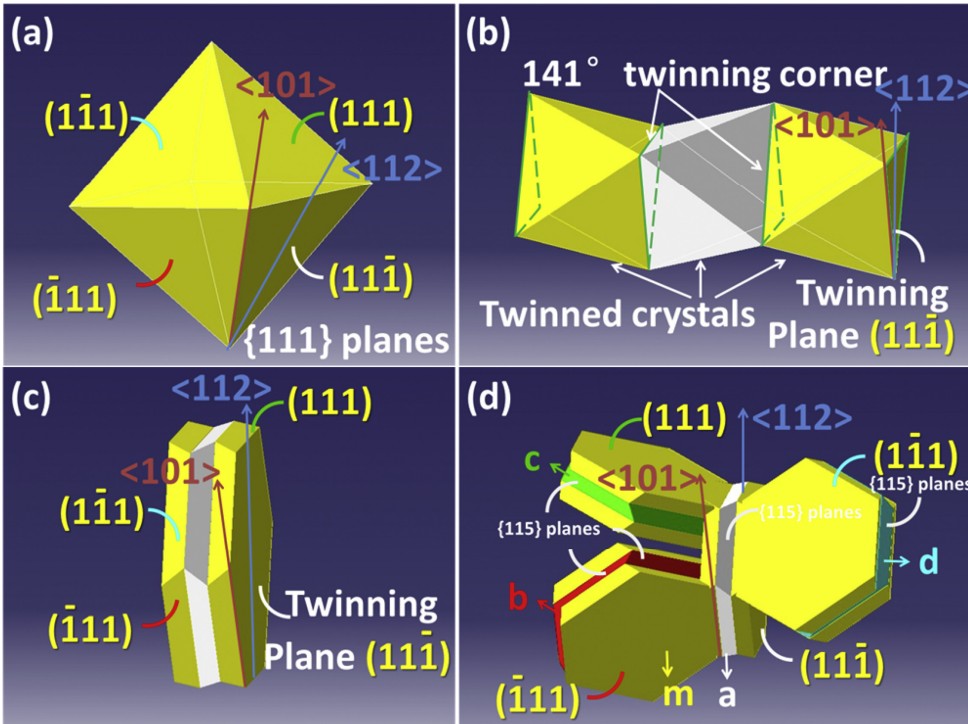

**Figure 3.** Octahedral configuration: (**a**) a faceted equiaxed grain m bounded by eight {111} planes; (**b**) occurrence of double parallel twin boundaries on the $(11\bar{1})$ plane of the crystal; (**c**) twinned lamellas grow on the $(11\bar{1})$ plane; (**d**) twinned lamellas grow on four different {111} planes [24]. Reprinted with permission from ref. [24]. Copyright 2018 Elsevier.

## 2.2. <111>-Oriented Tb-Dy-Fe Alloys Prepared by Directional Solidification in Magnetic Fields

Aiming to prepare Tb-Dy-Fe alloys with preferred orientation along <111> or close to <111>, researchers introduced a strong magnetic field to induce the crystal orientation in a specific direction during the directional solidification process. According to the relevant theory of crystal orientation induced by a magnetic field [25], if there is sufficient action time and rotation space, grains of materials with magnetocrystalline anisotropy will rotate and be oriented under the action of a strong magnetic field through Lorentz force, magnetic force and the magnetic moment of materials.

A high magnetic field in the horizontal direction was directly applied to the directional solidification process [26]. Liu et al. [27–31] obtained a higher <111> orientation and improved magnetostrictive properties by applying a constant magnetic field of 4.4 T during the solidification process of $Tb_{0.27}Dy_{0.73}Fe_{1.95}$ alloy, as shown in Figure 4. In addition, the best contrast of the domain image and the widest magnetic domain were obtained in the sample prepared under a 4.4 T magnetic field [27]. The required magnetic field in the range of 4–10 T to achieve <111> orientation increased with the increase in the cooling rate [28].

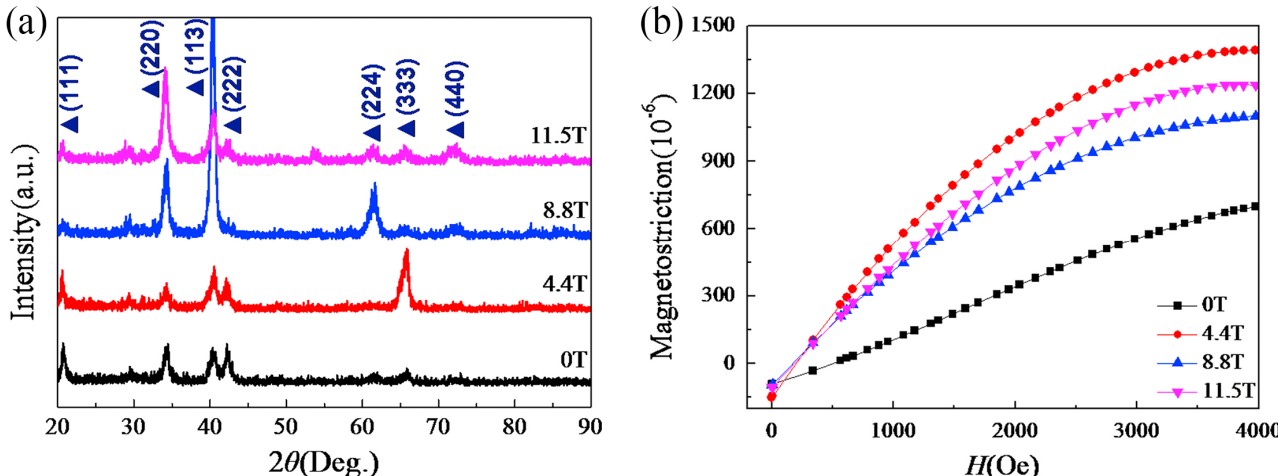

**Figure 4.** (**a**) XRD patterns in the plane perpendicular to the magnetic field direction and (**b**) magnetostriction for $Tb_{0.27}Dy_{0.73}Fe_{1.95}$ alloys solidified in various high magnetic fields [27]. Reprinted with permission from Ref. [27]. Copyright 2016 Elsevier.

Recently, researchers obtained both orientation and alignment along <111> by multiple magnetic field effects of the liquid phase and solid phase, as shown in Figure 5 [29,30]. In addition, magnetostrictive and mechanical properties were increased by adjusting the content, morphology and distribution of the (Tb, Dy)Fe₃ phase and WSP by coupling directional solidification with a high magnetic field [31].

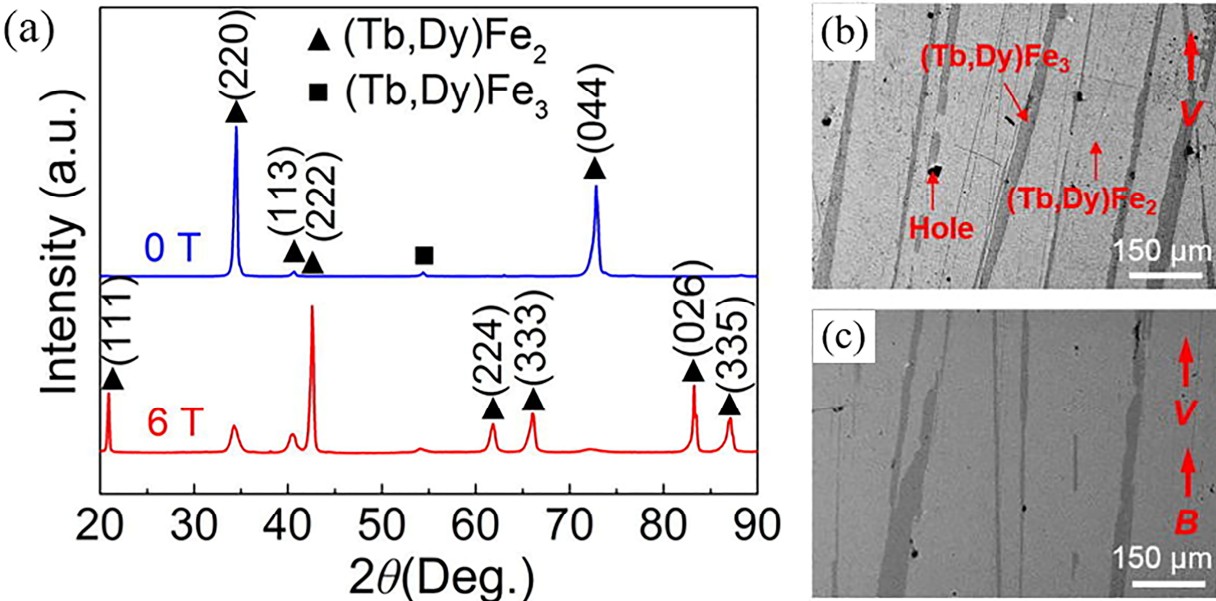

**Figure 5.** (**a**) XRD patterns of $Tb_{0.27}Dy_{0.73}Fe_{1.95}$ alloys on the transverse section; (**b**,**c**) SEM images of the alloy structures grown with (**b**) 0T and (**c**) 6 T magnetic field [29]. Reprinted with permission from ref. [29]. Copyright 2020 AIP Publishing.

Furthermore, when a gradient magnetic field strongly dependent on the cooling rate was applied during cooling and solidification, magnetic gradient $Tb_{0.27}Dy_{0.73}Fe_{1.95}$ alloy with gradient magnetostriction and saturation magnetization was obtained, as shown in Figure 6, which was attributed to the increase in the gradient of the orientation degree [32–34].

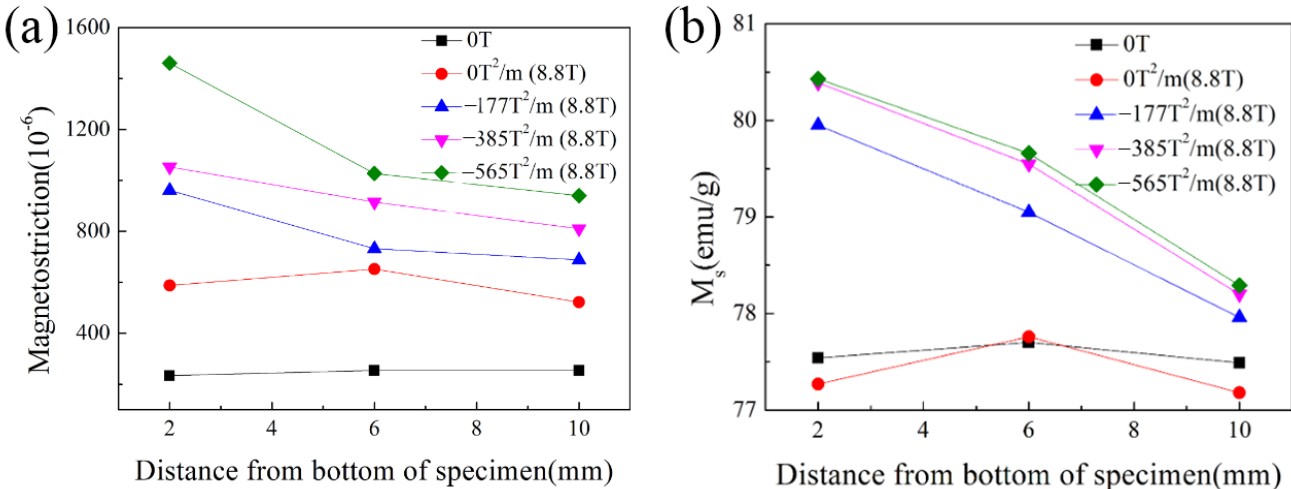

**Figure 6.** (**a**) The maximum magnetostriction of alloys solidified in various high magnetic fields at 4000 Oe and (**b**) saturation magnetization through the depths of alloys solidified in various high magnetic fields [32]. Reprinted with permission from Ref. [32]. Copyright 2016 World Scientific Publishing Company.

## 3. Effects of Substitute Elements on Magnetostriction of Tb-Dy-Fe Alloys

Cubic $RFe_2$ compounds usually have large magnetostriction at room temperature, but their magnetocrystalline anisotropy is also large. In this case, a strong external magnetic field is frequently required to achieve high performance in practical applications, which hinders the application of magnetostrictive $RFe_2$ materials. Initially, aiming to reduce magnetocrystalline anisotropy as well as maintain large magnetostriction, pseudobinary $RR'Fe_2$ compounds with an anisotropy compensation function were developed, that is, by alloying $RFe_2$ compounds with the same magnetostriction symbol but opposite anisotropy symbol. Furthermore, with a Laves phase structure, the magnetic properties of this alloy can be easily changed by various substitutions in rare-earth and 3d transition-metal sublattices.

### 3.1. Alloy System Containing Other Rare-Earth Elements

The introduction of the third rare-earth element can provide additional degrees of freedom for the $(Tb,Dy)Fe_2$ system, minimizing the first-order anisotropy constant $(K_1)$ and the second-order anisotropy constant $(K_2)$, which is considered as an approach to improve the magnetostriction of Tb-Dy-Fe alloys.

#### 3.1.1. Pr

$PrFe_2$ has a very high magnetostrictive coefficient (about 5600 ppm at 0 k), and the high Pr content is also conducive to anisotropic compensation. However, $Tb_{1-x}Pr_xFe_2$ is a noncubic phase when the Pr content exceeds 20% due to the large $Pr^{3+}$ radius, while high Pr content is conducive to anisotropic compensation [13]. During the last years, some studies have confirmed that $(Tb, Pr, Dy) Fe_2$ series compounds are an anisotropic compensation system [35–37].

One way to improve the performance of the $(Tb, Pr, Dy)Fe_2$ system is to replace Fe by adding Co, B and other elements. For instance, $(Tb_{0.7}Dy_{0.3})_{0.7}Pr_{0.3}(Fe_{1-x}Co_x)_{1.85}$ $(0 \leq x \leq 0.6)$ was composed of a $MgCu_2$-type C15 cubic Laves phase, with a small amount of a $PuNi_3$-type phase and rare-earth-rich phase [11]. The second phase of $Dy_{1-x}(Tb_{0.2}Pr_{0.8})_xFe_{1.93}$ $(0 \leq x \leq 0.5)$ obtained by atmospheric pressure annealing appeared when x exceeded 0.3 [37]. Shi et al. successively synthesized $Pr_xTb_{1-x}Fe_{1.9}$ $(0 \leq x \leq 1)$ [38], $Pr(Fe_{1-x}Co_x)_{1.9}$ $(0 \leq x \leq 0.5)$ [39], $Dy_{1-x}Pr_xFe_{1.9}$ $(0 \leq x \leq 1)$ [40] and $Pr_{1-x}Dy_x(Fe_{0.8}Co_{0.2})_{1.93}$ (x = 0.00, 0.05, 0.10, 0.20 and 0.30) [36] single cubic Laves compounds by high-pressure annealing. The magnetostriction of $Pr_{0.95}Dy_{0.05}(Fe_{0.8}Co_{0.2})_{1.93}$ alloy at 3kOe was 648 ppm, which is twice that of $Tb_{0.2}Dy_{0.58}Pr_{0.22}(Fe_{0.9}B_{0.1})_{1.93}$ (about 300 ppm) [35].

The $440_C$ XRD profiles and magnetostriction of $(Tb_{0.2}Pr_{0.8})_xDy_{1-x}Fe_{1.93}$ (x = 0.00, 0.05, 0.10, 0.20 and 0.30) single Laves phase compounds synthesized by high-pressure annealing are shown in Figure 7 [41]. Based on the consideration of anisotropy compensation and thermodynamic energy flattening, a rare-earth sublattice was designed, and the ternary composition phase diagram and the minimum anisotropy composition are shown in Figure 8 [42].

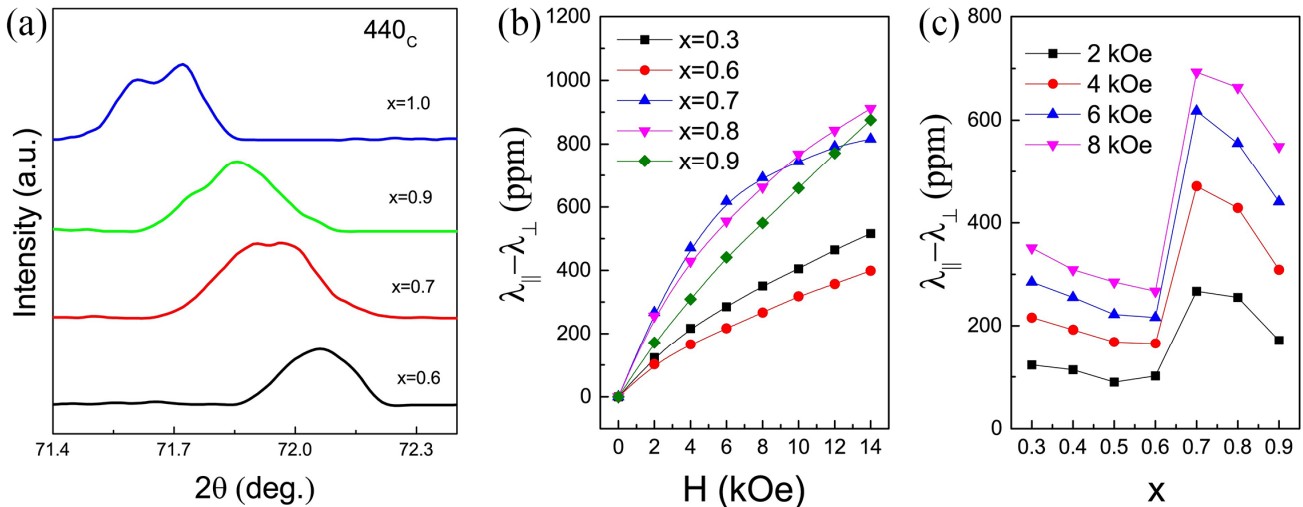

**Figure 7.** (**a**) Typical step-scanned $440_C$ XRD profiles of the $(Tb_{0.2}Pr_{0.8})_xDy_{1-x}Fe_{1.93}$ Laves phase, and room temperature magnetostriction $\lambda_{\parallel}-\lambda_{\perp}$ as a function of (**b**) the applied field and (**c**) the composition [41]. Reprinted with permission from Ref. [41]. Copyright 2019 Elsevier.

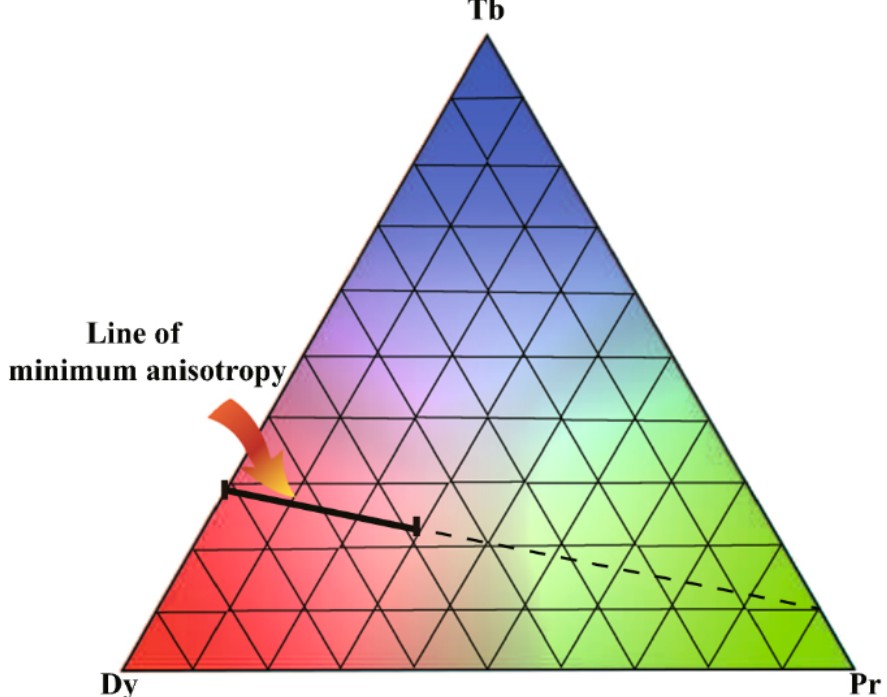

**Figure 8.** Designed ternary composition phase diagram for the Tb-Dy-Pr system: the location between two bias diagonals of the rare-earth composition is indicated along the expected minimum anisotropy line [42]. Reprinted with permission from Ref. [42]. Copyright 2020 Springer Nature.

### 3.1.2. Nd

Similar to Pr, many studies [43–52] have shown that the appropriate replacement of heavy rare-earth Tb and Dy with light rare-earth Nd is an effective approach to decrease magnetocrystalline anisotropy and improve magnetostriction. For example, the $Tb_{0.4}Dy_{0.5}Nd_{0.1}Fe_{1.303}$ compound possessed the large magnetostriction value of 1300 ppm at 5 kOe [47]. Pan et al. [45,48] obtained good magnetoelasticity in $Tb_{0.3}Dy_{0.6}Nd_{0.1}(Fe_{0.8}Co_{0.2})_{1.93}$ by improving the annealing process.

$Tb_{0.2}Nd_{0.8}(Fe_{0.8}Co_{0.2})_{1.9}$ ribbons with high Nd content were prepared by melt spinning and low-temperature annealing, which provided another effective method for the synthesis of a C15 cubic Laves phase structure with high Nd content [49]. The results indicated that a higher solidification rate is conducive to the elimination of the $(Tb, Nd)Fe_3$ phase. A single cubic Laves phase with a <111> easy magnetization direction at room temperature was obtained at 45 m/s runner speed and 773 K annealing temperature [50]. Subsequently, a series of $Tb_{0.2}Nd_{0.8}(Fe_{1-x}Co_x)_{1.9}$ ($0 \leq x \leq 0.4$) compounds were fabricated by rapid melt quenching in order to study the effects of Co substitution for Fe. It was found that the lattice parameter decreased with increasing x, and the $(\lambda_\parallel-\lambda_\perp)$ of ribbons with x = 0.1–0.2 at 10 kOe was 306 ppm and 321 ppm, respectively [51]. That is, an appropriate amount of Co instead of Fe could promote the formation of a single Laves phase and improve the magnetostrictive properties of $Tb_{0.2}Nd_{0.8}(Fe_{1-x}Co_x)_{1.9}$ ribbons.

Recently, the spin configuration phase diagram (Figure 9a) of $Tb_{0.27}Dy_{0.73-x}Nd_xFe_2$ ($0 \leq x \leq 0.4$) was designed based on the experimental results of magnetization with varying temperature, magnetic susceptibility and XRD analysis [52]. In this case, it was found that the Curie temperature, spin reorientation temperature, saturation magnetization and magnetostriction (Figure 9b) of the $Tb_{0.27}Dy_{0.73-x}Nd_xFe_2$ compound decreased with the increase in Nd concentration. Moreover, a value of $\lambda_{111}$ = 1700 ppm in the $Tb_{0.27}Dy_{0.63}Nd_{0.1}Fe_2$ Laves phase compound was obtained by high-pressure annealing.

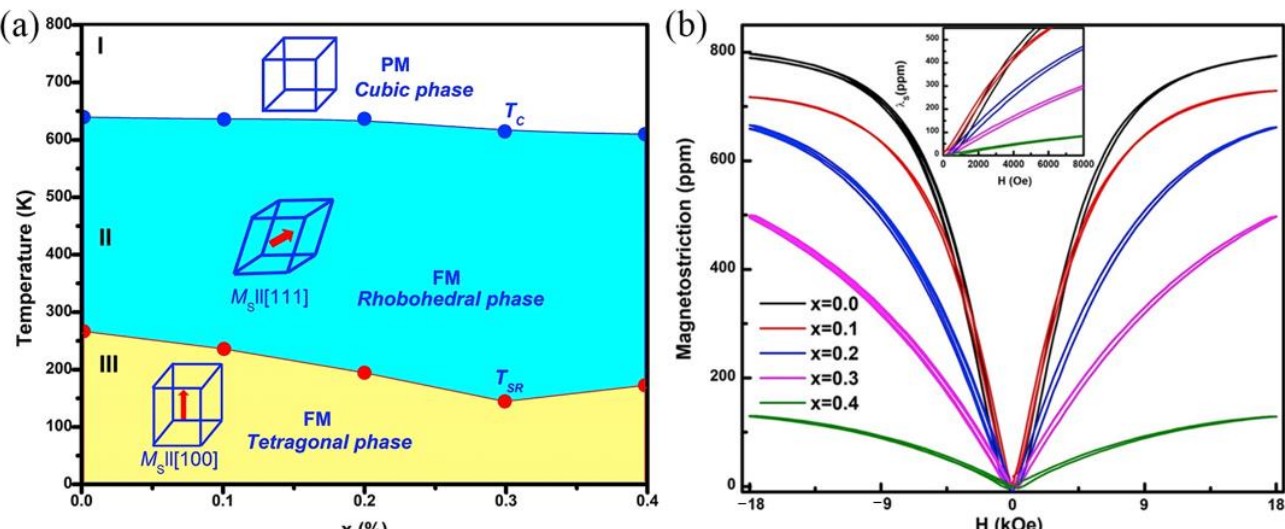

**Figure 9.** For $Tb_{0.27}Dy_{0.73-x}Nd_xFe_2$: (**a**) spin configuration phase diagram accompanied by the cubic, rhombohedral and tetragonal crystal symmetries and (**b**) room temperature magnetostriction ($\lambda_S$) (x = 0.0, 0.1.0.2, 0.3, 0.4) [52]. Reprinted with permission from ref. [52]. Copyright 2020 Elsevier.

### 3.1.3. Ho

Substituting a small amount of Ho for Tb and Dy in $Tb_{0.3}Dy_{0.7}Fe_2$ can significantly decrease anisotropy and hysteresis loss [53]. Moreover, the addition of Ho also narrowed the temperature range between the liquidus temperature and the peritectic temperature, which was beneficial for reducing the pre-peritectic of the $(Tb, Dy, Ho)Fe_3$ phase. $Tb_{0.26}Dy_{0.49}Ho_{0.25}Fe_{1.9}$ had both large magnetostriction and small hysteresis under a low magnetic field. Furthermore, it has also been found that magnetic annealing effectively

increases magnetostriction [54]. Figure 10 shows the anisotropic compensation effects of Ho and Pr in $Tb_{0.1}Ho_{0.9-x}Pr_x(Fe_{0.8}Co_{0.2})_{1.93}$ alloys and the effects of their components on the phase structure and magnetostriction [55].

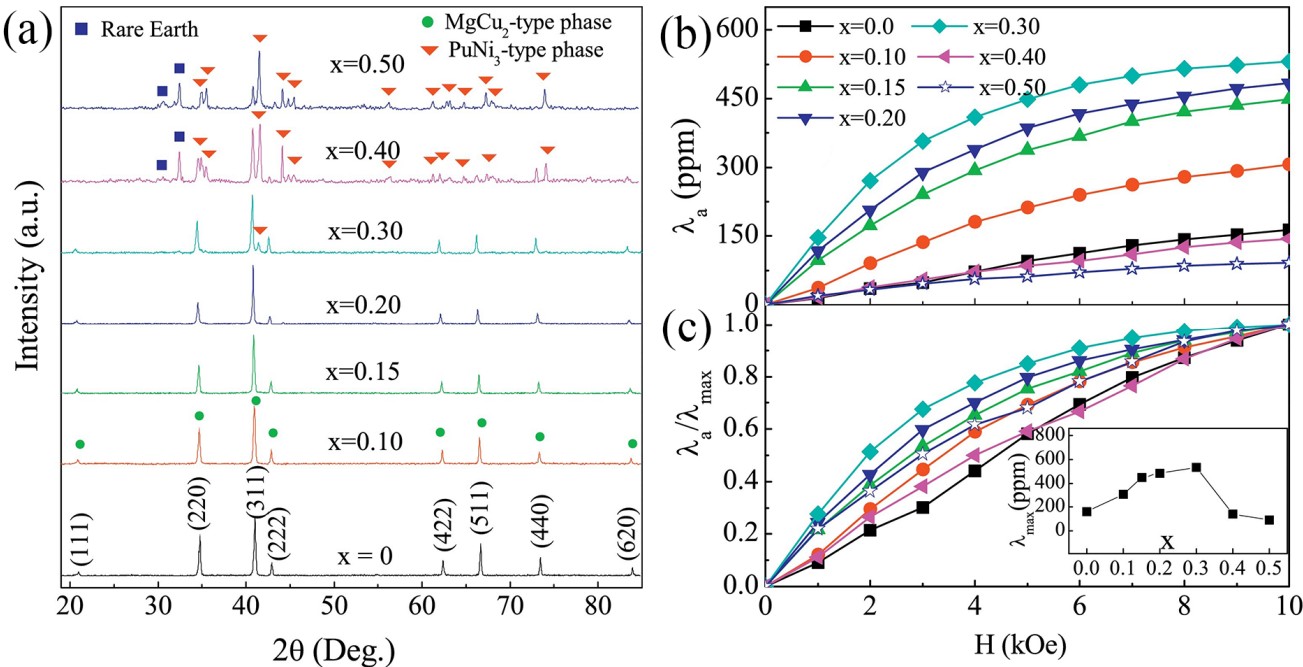

**Figure 10.** (**a**) XRD patterns and magnetic-field dependence of (**b**) magnetostriction and (**c**) normalized magnetostriction for $Tb_{0.1}Ho_{0.9-x}Pr_x(Fe_{0.8}Co_{0.2})_{1.93}$ alloys [55]. Reprinted with permission from Ref. [55]. Copyright 2016 Elsevier.

Moreover, the addition of Ho reduced the saturated magnetic field (220 kA/m) and dynamic magnetic loss of Tb-Dy-Fe fiber composites [56], and the maximum magnetostriction of the Tb-Dy-Ho-Fe/epoxy composite was 695 ppm when the Ho content x = 0.31 [57].

### 3.2. Alloy System Containing Other Elements

In a system containing light rare-earth elements, an appropriate amount of Co instead of Fe promoted and stabilized the formation of the cubic Laves phase [51,58,59]. In addition, 20 at% Co instead of Fe was found to increase the Curie temperature and saturation magnetization and improve the magnetostriction of $Tb_{0.2}Nd_{0.8}Fe_{1.9}$ ribbons [51]. Co substitution for Fe was able to extend the operating temperature range for $Tb_{0.36}Dy_{0.64}Fe_2$ by increasing the Curie temperature ($T_c$) or decreasing the spin reorientation temperature ($T_r$) [60,61]. Recently, Yang et al. found a new "Griffiths-like transition" in $Tb_{0.3}Dy_{0.7}(Co_{1-x}Fe_x)_2$ when x < 0.8 and suggested that its disappearance was due to the interaction between Fe and Co [62]. In addition, enhanced magnetostriction (818 ppm) and high $T_c$ (707 K) were obtained at x = 0.8. Theoretically, the doping of transition metals modulated the exchange action between 3d-3d and 3d-4f atoms, and it is expected to improve the elastic energy and magnetostatic energy.

Based on the different effects of the addition of elements on properties, Wang et al. [59] divided different elements into two types: those that readily formed phases with rare-earth elements and those that readily formed phases with the enthalpy of mixing between atomic pairs. After adding Nb, Ti and V elements, the second phases $NbFe_2$, $Fe_2Ti$ and FeV, respectively, were dispersed in the Tb-Dy-Fe matrix alloy, as shown in Figure 11, which can inhibit the formation of the harmful $RFe_3$ phase.

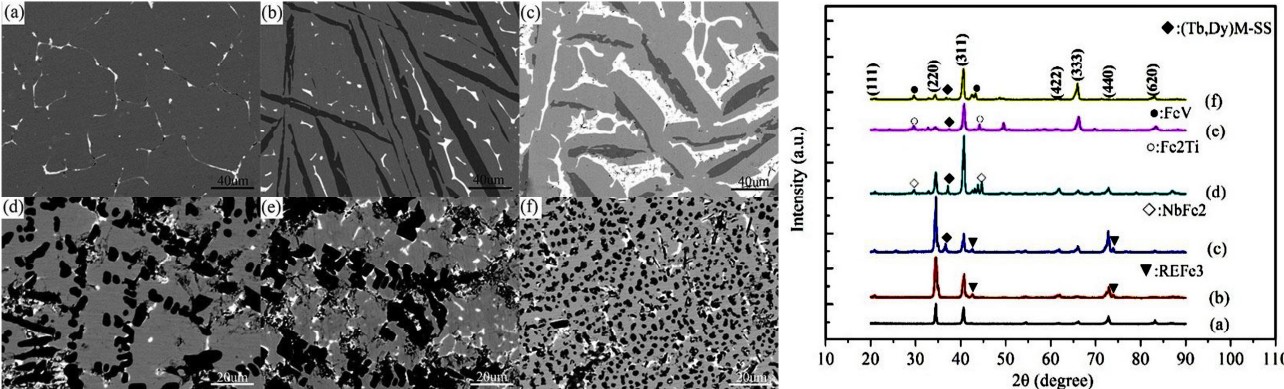

**Figure 11.** BSEM images and X-ray diffraction patterns of the alloys (**a**) $Tb_{0.3}Dy_{0.7}Fe_2$; (**b**) $(Tb_{0.3}Dy_{0.7})_{0.3}$ $Fe_{0.6}Co_{0.1}$; (**c**) $(Tb_{0.3}Dy_{0.7})_{0.3}Fe_{0.6}Cu_{0.1}$; (**d**) $(Tb_{0.3}Dy_{0.7})_{0.3}Fe_{0.6}Nb_{0.1}$; (**e**) $(Tb_{0.3}Dy_{0.7})_{0.3}Fe_{0.6}Ti_{0.1}$; (**f**) $(Tb_{0.3}Dy_{0.7})_{0.3}Fe_{0.6}V_{0.1}$ [59]. Reprinted with permission from ref. [59]. Copyright 2018 Elsevier.

## 4. Mechanical Properties of Tb-Dy-Fe Alloys

The $MgCu_2$-type $RFe_2$ phase in the alloy provides a large magnetostrictive coefficient, but at the same time, the Laves phase is brittle and fractures easily at room temperature because of its topological close packing structure and lack of available slip system. There are also a large number of parallel acicular Widmanstatten precipitates (WSPs) in the Laves phase, that is, tiny $(Tb, Dy)Fe_3$ phases [63,64]. The complex staggered distribution of WSPs and the lamellar distribution of $(Tb, Dy)Fe_3$ phases in the matrix will adversely affect the mechanical properties.

As the Fe content decreases from a stoichiometric ratio of 2.0, the strength increases significantly, because the ductile rare-earth phase serving as the skeleton network delays crack propagation in the brittle matrix [65]. Heat treatment [6,66] can also improve the mechanical properties by controlling the dispersion and uniform distribution of the spherical rare-earth-rich phase.

In studies on alloying, the $NbFe_2$ phase [67] and $(Tb, Dy)Cu$ phase [68] were found to have the ability to prevent crack propagation, which was beneficial for the improvement of the mechanical properties of the Tb-Dy-Fe alloys. It was suggested that the existence of a soft phase in the alloy made the material exhibit inelastic strain under tensile or shear load. For instance, the soft $(Tb, Dy)Cu$ phase played a key role in stopping or changing the direction of cracks, as shown in Figure 12. The addition of Cu increased the fracture toughness by 2–3 times, and the alloy with 1 at% Cu showed the best fracture toughness of 3.47 MPa·m$^{1/2}$. With the addition of Nb, the fracture toughness was 1.5–5 times higher than that of Nb-free alloy.

Inspired by the ductility $(Tb, Dy)Cu$ phase, the low-melting-point Dy-Cu alloy was introduced to the grain boundary phase of directionally solidified $Tb_{0.3}Dy_{0.7}Fe_{1.95}$ alloy by grain boundary diffusion [69]. The results revealed that the magnetostrictive properties were maintained at 1021–1448 ppm, and the optimum bending strength was increased by nearly 2.6 times (Figure 13). The large magnetostriction was attributed to the matrix Laves phase structure and stable preferred orientation during the grain boundary diffusion process. In addition, the grain boundary phase was mainly composed of the ductile $(Dy,Tb)Cu$ phase, which can retard crack propagation and improve the mechanical strength of $Tb_{0.3}Dy_{0.7}Fe_{1.95}$ alloy.

The content, morphology and distribution of the $(Tb, Dy)Fe_3$ phase and WSP in $Tb_{0.3}Dy_{0.7}Fe_{1.95}$ alloy can be controlled by the effect of a magnetic field on grain orientation and element diffusion during the solidification process [31]. When the angle between the $(Tb, Dy)Fe_3$ phase and the grain growth direction was the smallest and the WSP content was low, the alloy had better mechanical properties. The application of a 6T magnetic field during the solidification process improved the mechanical properties of the alloy at the same growth rate, as shown in Figure 14.

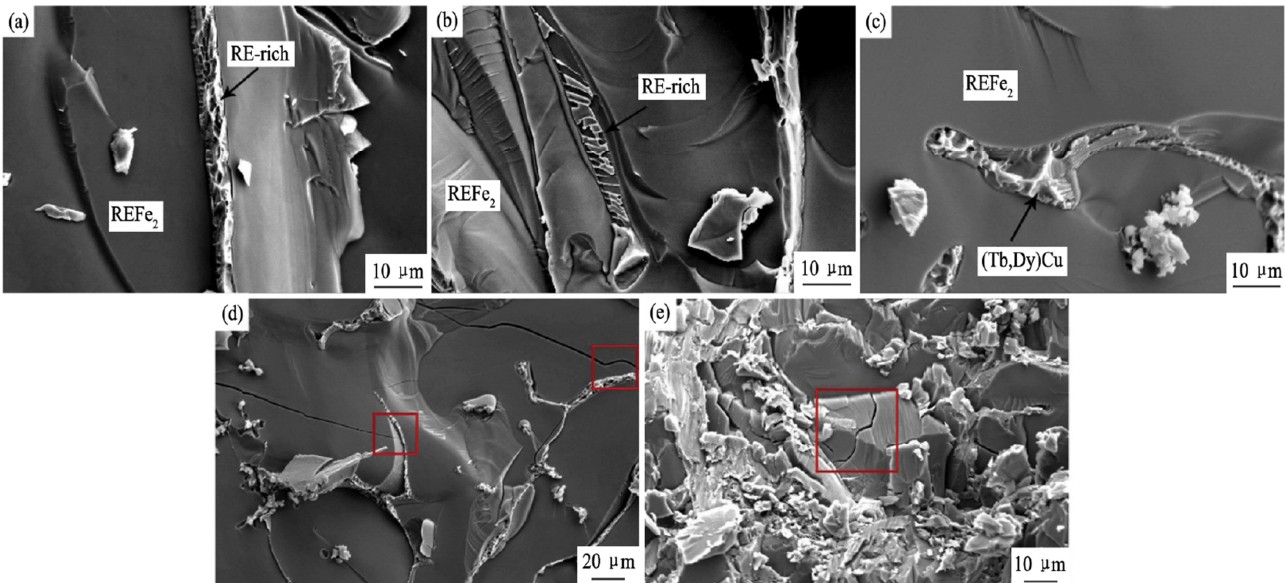

**Figure 12.** SEM images of crack propagation and soft phase deformation during the compressive fracture process in alloys. (**a**,**b**) $(Tb_{0.3}Dy_{0.7})_{0.37}Fe_{0.63}$; (**c**,**d**) $(Tb_{0.3}Dy_{0.7})_{0.37}Fe_{0.62}Cu_{0.01}$; (**e**) $(Tb_{0.3}Dy_{0.7})_{0.37}Fe_{0.53}Cu_{0.1}$ [68]. Reprinted with permission from ref. [68]. Copyright 2019 Elsevier.

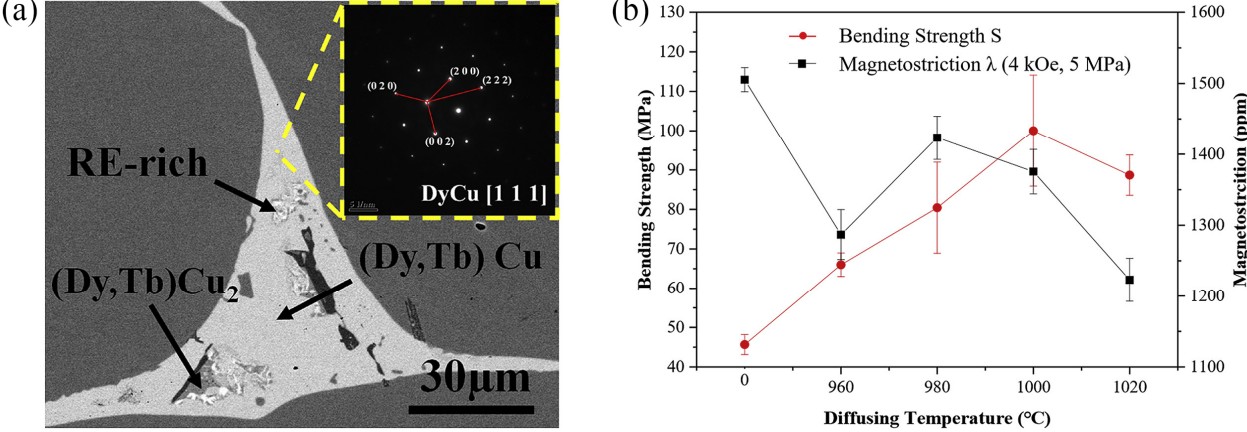

**Figure 13.** (**a**) SEM images and (**b**) bending strength and magnetostrictive properties of the $Tb_{0.3}Dy_{0.7}Fe_{1.95}$ alloy diffused by $DyCu_2$ alloy at 980 °C for 3 h, followed by quenching to room temperature; the inset in (**a**) is SAED pattern of DyCu along the [111] zone axis [69]. Reprinted with permission from ref. [69]. Copyright 2020 Elsevier.

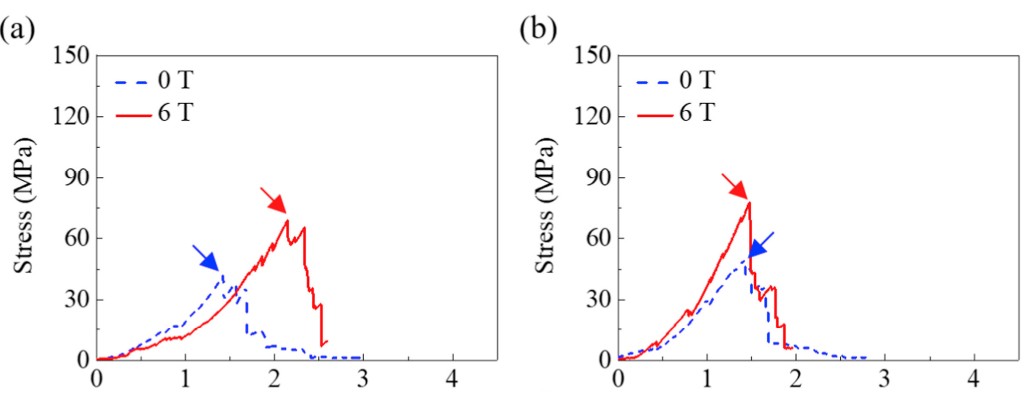

**Figure 14.** *Cont.*

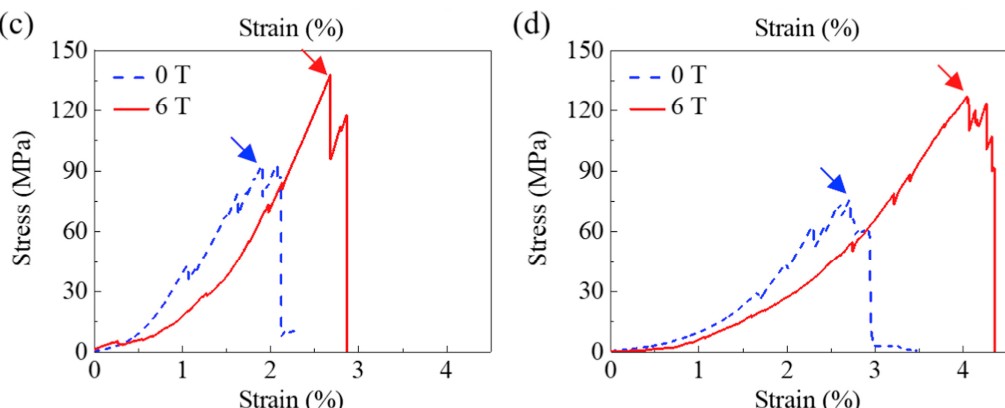

**Figure 14.** Stress–strain curves of $Tb_{0.27}Dy_{0.73}Fe_{1.95}$ alloys that were solidified directionally at various growth velocities without and with a 6 T magnetic field: (**a**) 25 μm/s; (**b**) 50 μm/s; (**c**) 100 μm/s; (**d**) 200 μm/s [31]. Reprinted with permission from ref. [31]. Copyright 2020 Elsevier.

## 5. Structural Origin and Magnetic Morphotropic Phase Boundary (MPB) of Tb-Dy-Fe Alloys

To explore the more efficient Tb-Dy-Fe giant magnetostrictive material system, it is necessary to understand the basic principle of magnetostriction, rather than only having a phenomenological understanding. During the last years, with the development of high-resolution synchrotron radiation X-ray diffraction technology, it has been increasingly recognized that the ferromagnetic phase transition process is accompanied by a change in structural symmetry. Significantly, in-depth research of the magnetic morphotropic phase boundary (MPB) has been carried out. In previous research, the synchrotron radiation X-ray diffraction data of $Tb_{0.3}Dy_{0.7}Fe_2$ at 300 K revealed that the {440} and {222} reflections were obviously split, reflecting rhombic symmetry and lattice stretching along the [111] direction. Structurally, this was the process of spin redirection from <001> to <111>, that is, the transition from the *T* phase with small lattice distortion along the <001> direction to the *R* phase with large lattice elongation along the <111> direction [70].

The phase diagram of $Tb_{1-x}Dy_xFe_2$ (Figure 15) was obtained through magnetometry and synchrotron XRD experiments [71–73]. It indicates that ferromagnetic MPB is composed of two crystal structures of the parent compounds $TbFe_2$ and $DyFe_2$, with a broadening MPB width at higher temperatures. Furthermore, a simulation based on the energy model demonstrated that the exchange energy narrowed the MPB region by affecting the magnetic phase transition process. This could also be used to explain the above abnormal phenomenon [72]. The exchange interaction was weakened with the increase in temperature, which corresponded to the broadening of the MPB region. In particular, the best point of magnetomechanical application was not centered on MPB but on one side of the rhombohedron. In addition, this local rhombohedral symmetry was further proved by high-resolution transmission electron microscopy. The local nanodomains of ferromagnetic rhombohedral and tetragonal phases coexist in $Tb_{0.3}Dy_{0.7}Fe_2$, as shown in Figure 16 [74]. This is similar to the hierarchical nanodomain structure in ferroelectric materials.

The local stress environment generated by these randomly distributed tetragonal nanodomains in the rhombohedral matrix affected the interaction between $Fe_1$ and $Fe_2$ atoms and caused anomalies in the lattice, as shown in Figure 17 [18]. The weak $Fe_1$-$Fe_2$ bond was sensitive to the environment, which played an important role in the lattice characteristics of the rhombohedral phase. As a result, the lattice became more orderly with the increase in Dy and more stable Dy-rich phase, which was confirmed by X-ray absorption spectroscopy (XAS) techniques (Figure 18) [18].

These studies also enriched the understanding of the magnetic structure and the origin of large magnetostriction of Tb-Dy-Fe alloys. In detail, the maximum magnetostriction was related to the transition from the *T* phase with a smaller lattice along the <001> direction to the *R* phase with larger lattice elongation along the <111> direction.

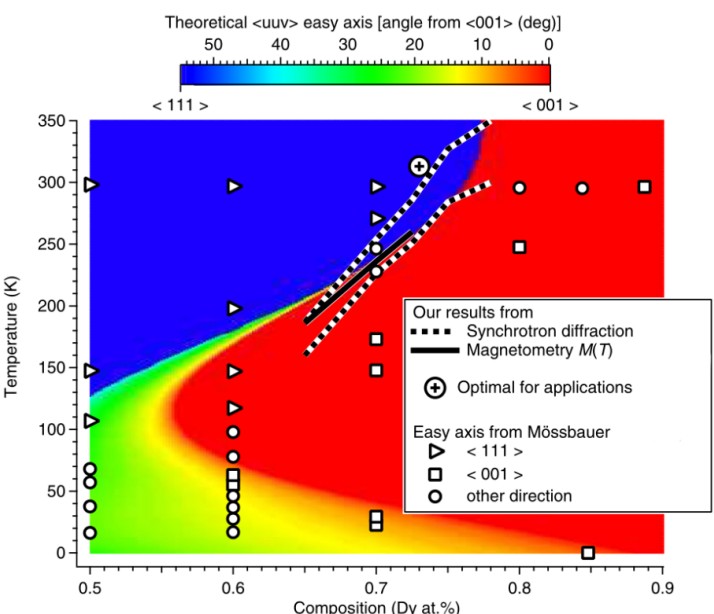

**Figure 15.** Phase diagram of Tb$_{1-x}$Dy$_x$Fe$_2$. The background shading shows the magnetic easy axis direction calculated using anisotropy parameters from crystal field theory (see text) [71]. Overlayed is the morphotropic phase boundary determined from our synchrotron XRD (dotted lines) and magnetometry (solid line) measurements, as well as the easy axes reported previously on the basis of Mössbauer spectroscopy (open symbols) [73]. A cross in a circle indicates the optimal temperature (40 °C) for magnetomechanical device applications, as determined for Tb$_{1-x}$Dy$_x$Fe$_2$ x = 0.73 [2].

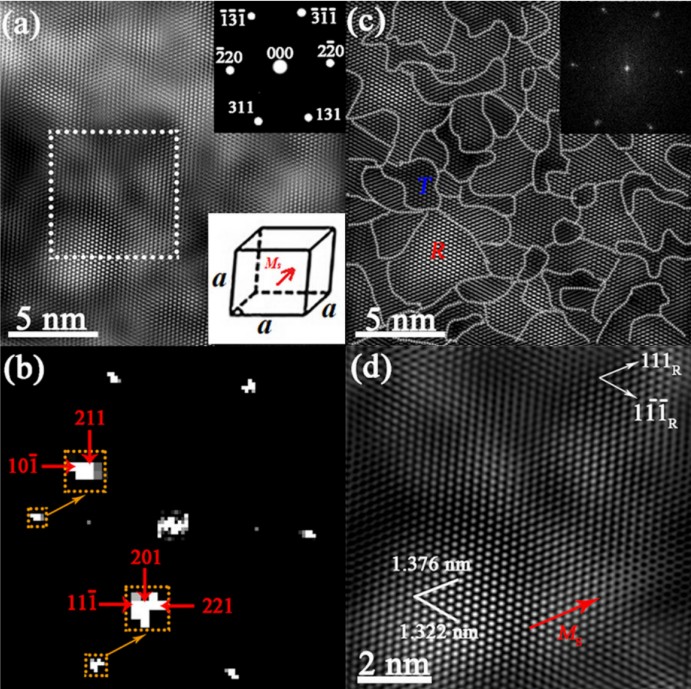

**Figure 16.** Domains of Tb$_{0.3}$Dy$_{0.7}$Fe$_2$ at 293 K revealed by HRTEM [74]. (**a**) HRTEM image taken along the [1$\bar{1}$4]C incident direction. Upper inset is the SAED pattern, and bottom inset is the schematic unit cell with spontaneous magnetization direction. (**b**) FFT of the white rectangle in (**a**) shows splitting reflection spots due to rhombohedral lattice distortion. (**c**) IFFT image by using the {311}C/{111}R reflections in the bottom inset of (**a**), corresponding to the same area in (**a**). The inset is the corresponding FFT spectrum. (**d**) IFFT image by using the {311}C/{111}R reflections from the FFT in (**b**). Adapted with permission from ref. [74]. Copyright 2014 Elsevier.

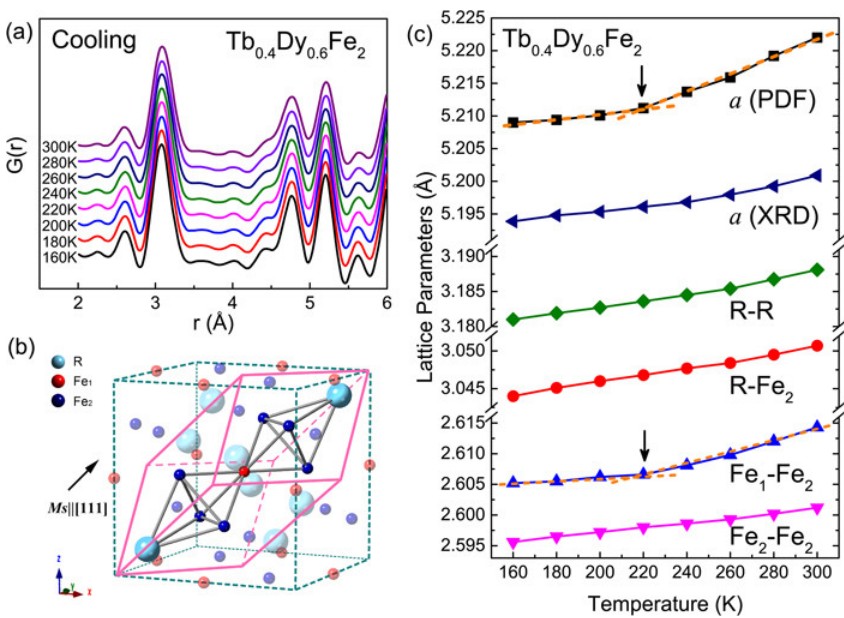

**Figure 17.** (**a**) Temperature-dependent pair distribution function (PDF) for $Tb_{0.4}Dy_{0.6}Fe_2$; (**b**) the rhombohedral cell (pink solid) and cubic cell (teal dashed) for $Tb_{1-x}Dy_xFe_2$; (**c**) temperature-dependent lattice constants and bond lengths for $Tb_{0.4}Dy_{0.6}Fe_2$ [18]. Reprinted with permission from ref. [18]. Copyright 2020 AIP Publishing.

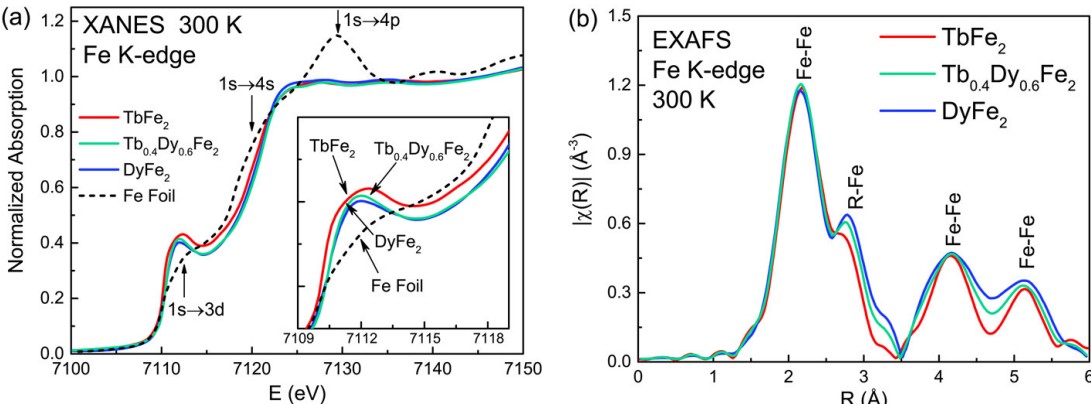

**Figure 18.** (**a**) Room temperature Fe K-edge XANES spectra for Fe foil and $Tb_{1-x}Dy_xFe_2$ samples. (**b**) Room temperature Fe K-edge EXAFS spectra for $Tb_{1-x}Dy_xFe_2$ [18]. Reprinted with permission from ref. [18]. Copyright 2020 AIP Publishing.

It was proposed that the change in the magnetization direction under the action of an external magnetic field could be realized by field-preferred domain growth, which could contribute to explaining large magnetostriction in the low field [75]. The comparison of the diffraction peak intensity of $TbFe_2$ and $Tb_{0.4}Dy_{0.6}Fe_2$ compounds is shown in Figure 19. This magnetostriction of $Tb_{0.4}Dy_{0.6}Fe_2$ was considered to be related to the reduction in rhombohedral distortion caused by replacing Tb by Dy, resulting in field-induced domain conversion, which was more sensitive to the external field [75].

The domain structure and transition near ferromagnetic MPB are also of great significance to understanding the large magnetostriction for Tb-Dy-Fe alloys. The micro-mechanism of domain strain behavior near ferromagnetic MPB was intuitively illustrated by the phase-field method, combining micromagnetic and micro-elastic theory. This large magnetostrictive strain was considered to be due to the low-energy rotation path of the local magnetization vector in the phase coexistence region. In particular, the tetragonal phase as the intermediate phase provided a low-energy rotation channel for the diamond

phase domain from other directions to the external field direction [76,77]. Similar to ferroelectrics, it was believed that the flattening of thermodynamic energy should lead to the sensitive response of the ferromagnetic phase. Through the introduction of $Tb_{0.1}Pr_{0.9}$ to Co-doped $Tb_{0.27}Dy_{0.73}Fe_2$ alloy, Hu et al. explored the feasibility of this assumption through phase-field simulation [42]. They provided a feasible strategy for the design of an ultrasensitive magnetostrictive response with minor metastable orthorhombic phases as bridging domains, as shown in Figure 20.

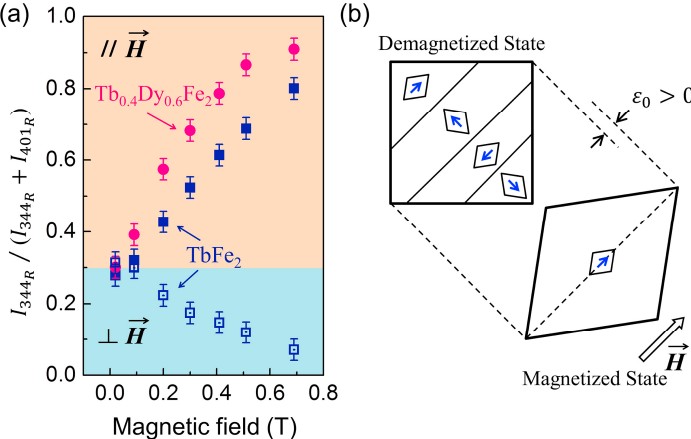

**Figure 19.** (**a**) Integrated intensity fraction of $(344)_R$ diffraction peak and (**b**) schematic of the microstructural evolutions in $TbFe_2$ and $Tb_{0.4}Dy_{0.6}Fe_2$ compounds under an applied magnetic field [75]. Reprinted with permission from ref. [75]. Copyright 2016 Elsevier.

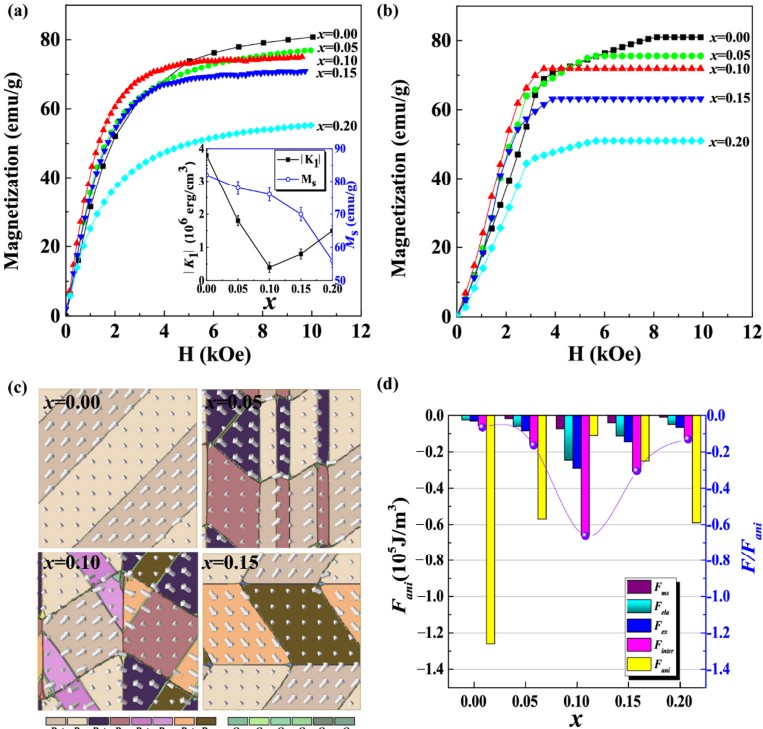

**Figure 20.** (**a**) Magnetization curves for $(Tb_{0.27}Dy_{0.73})_{1-x}(Tb_{0.1}Pr_{0.9})_x(Fe_{0.9}Co_{0.1})_2$ compounds ($0 \leq x \leq 0.2$). The inset shows the absolute values of the first-order anisotropy constant $|K_1|$ and saturation magnetization $M_S$ at room temperature. (**b**) The calculated average magnetization $M_{xy}$ from phase-field simulation for the corresponding experimental compositions. (**c**) Snapshots of domain structures for the samples with $0.00 \leq x \leq 0.15$. (**d**) Energy analysis for the samples with $0.00 \leq x \leq 0.20$ [42]. Reprinted with permission from ref. [42]. Copyright 2020 Springer Nature.

The optimum Dy content of x = 0.73 at room temperature was verified in $Tb_{1-x}Dy_xFe_2$ by first-principles calculations [78]. Combining first-principles calculations with the crystal-field approach, the critical Dy concentration was 0.78, and the corresponding magnetostrictive coefficient $\lambda_{111}$ was 2700 ppm. The calculated spin-orientation diagram reproduced the experimental results for the [111] and [100] easy directions, as shown in Figure 21.

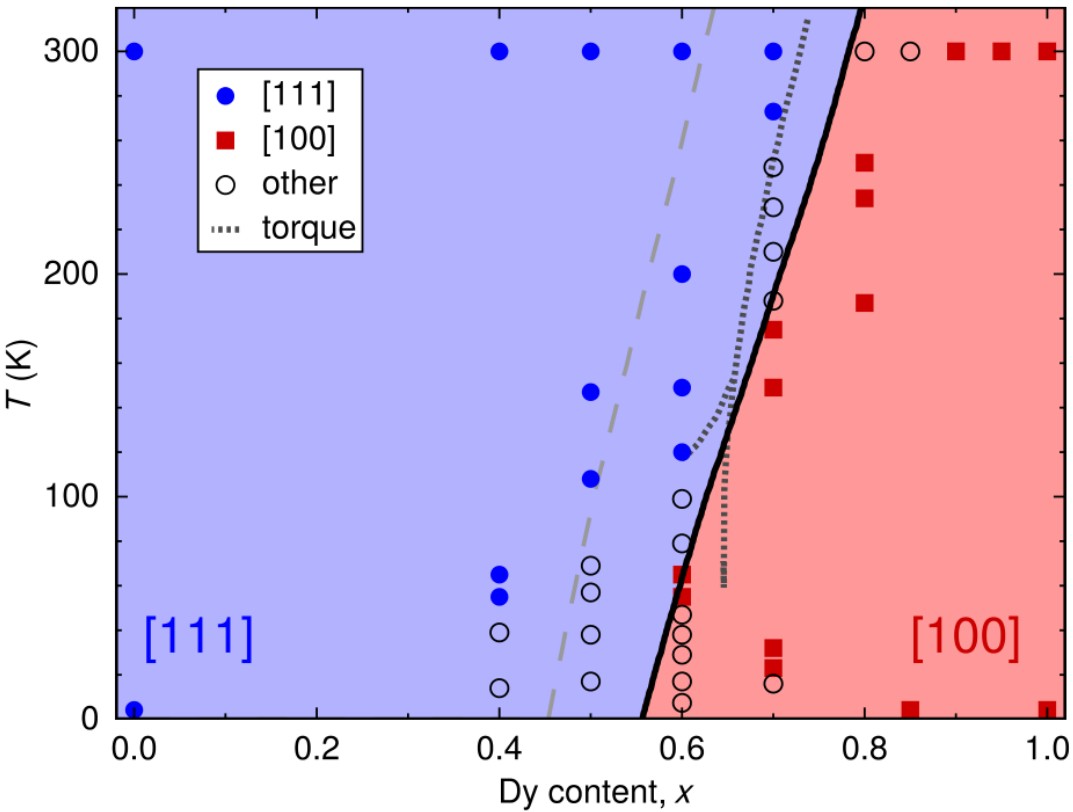

**Figure 21.** The easy direction of magnetization of $Tb_{1-x}Dy_xFe_2$, calculated by minimizing $E(\hat{u}, \varepsilon, x, T)$ (red- and blue-shaded regions) [78]. Reprinted with permission from ref. [78]. Copyright 2020 American Physical Society.

## 6. Progress on Tb-Dy-Fe Giant Magnetostrictive Composites

### 6.1. Polymer-Banded Tb-Dy-Fe Composite

Based on the requirements of device design, in recent years, some research has focused on giant magnetostrictive polymer composites (GMPCs) with high resistivity, a wide frequency response range and ease of formability. In a GMPC, Tb-Dy-Fe alloy particles were banded in the resin matrix in the form of particles or fibers. Tb-Dy-Fe alloy particles could be randomly dispersed (0–3 type) or arranged in a chain (1–3 type or pseudo-1–3 type) by applying different preparation methods [79]. The size, volume fraction and crystallographic orientation [80,81] of Tb-Dy-Fe alloy particles, as well as the bond strength [82] between the matrix and particles, will affect the magnetostrictive properties of the composite. After compounding, the eddy current was greatly reduced [83], which was conducive to the improvement of dynamic properties, and the mechanical properties were also greatly improved. The main characteristics of GMPCs of several Tb-Dy-Fe systems reported in the literature are summarized in Table 1.

**Table 1.** Properties of various GMPCs reported in the literature.

| Composite | Orientation | Preparation | Magnetostrictive Particle Morphology | Magnetostrictive Particle Size | Particle Content | Magnetostriction or Comments | References |
|---|---|---|---|---|---|---|---|
| $Tb_{0.3}Dy_{0.7}Fe_{1.9}$/epoxy | <111> | 8000 Oe magnetic field curing | Particles; pseudo-1–3 chain structure | >300 μm | 40 vol% | 1358 ppm (at 17 MPa) | [84] |
| $Tb_{0.3}Dy_{0.7}Fe_{1.92}$/epoxy | - | 8000 Oe magnetic field curing | - | 200–300 μm | 40 vol% | Cut-off frequency is 6800 kHz; loss factor is only 4.3% of that for the monolithic Tb-Dy-Fe alloy (at 10 kHz and 10 mT) | [83] |
| $Tb_{0.4}Dy_{0.5}Nd_{0.1}(Fe_{0.8}Co_{0.2})_{1.93}$/epoxy | <111> | 10 kOe magnetic field curing | Particles; pseudo-1–3 chain structure | ≤150 μm | 20 vol% | 390 ppm ($\lambda_a$ is 650 ppm at 6 kOe) | [45] |
| Terfenol-D/epoxy | <112> | 1885 Oe magnetic field curing | Powder, particle; pseudo-1–3 chain structure | 5–300 μm | 70 vol% | 720 ppm (at 9 MPa) | [85] |
| $(Tb_{0.15}Ho_{0.85}Fe_{1.9})_{0.31}$ + $(Tb_{0.3}Dy_{0.7}Fe_{1.9})_{0.69}$/epoxy | | Pressure curing molding | Particles; pseudo-1–3 chain structure | 75–180 μm | 94 wt% | 605 ppm | [57] |
| $Tb_{0.25}Dy_{0.45}Ho_{0.30}Fe_{1.9}$/epoxy | <110> | 120 °C bonding molding | <110> staple fiber | 0.8 mm × 0.8 mm ×12 mm | 90 vol% | 220 kA/M saturated magnetic field; 5 kA/M coercivity; the total loss at 20 kHz is 115 W/m$^3$ | [56] |
| $Tb_{0.2}Dy_{0.55}Pr_{0.25}(Fe_{0.8}Co_{0.2})_{1.93}$/epoxy | <110> | 8042 Oe magnetic field curing | Particles; pseudo-1–3 chain structure | 75–150 μm | 30 vol% | 110 ppm ($\lambda_{||}$, at 80 kA/m); 580 ppm ($\lambda_a$, at 950 kA/m) | [86] |
| $Tb_{0.5}Dy_{0.5}Fe_{1.95}$/epoxy | <111> | Two-step method with 10 kOe dynamic magnetic orientation | Lamellar structure | 100–200 μm | 57 vol% | 1500 ppm | [87] |
| $Tb_xDy_{0.7-x}Pr_{0.3}(Fe_{0.9}B_{0.1})_{1.93}$/epoxy | <111> | 8042 Oe magnetic field curing | Particles; pseudo-1–3 chain structure | 60–150 μm | 30 vol% | $d_{33}$~2.2 nm/A (Hbias~80 kA/m) | [88] |

Recently, GMPCs with a layered structure were prepared by dynamic orientation in an oscillating magnetic field [89]. Jiang et al. [87] further obtained a Tb-Dy-Fe/epoxy particle composite with a high alloy particle volume fraction (57%) and high saturation magnetostriction (1500 ppm) by using a two-step dynamic orientation method, as shown in Figure 22b, and the energy density shown in Figure 22d was markedly improved. From Figure 23, it can be seen that the Tb-Dy-Fe alloy particles were first dynamically magnetically oriented in the liquid epoxy resin and then molded and concentrated in the horizontal magnetic field to remove the excess resin before curing. In addition, the defect-free matrix and anisotropic layered structure prepared by this method can effectively transfer the strain.

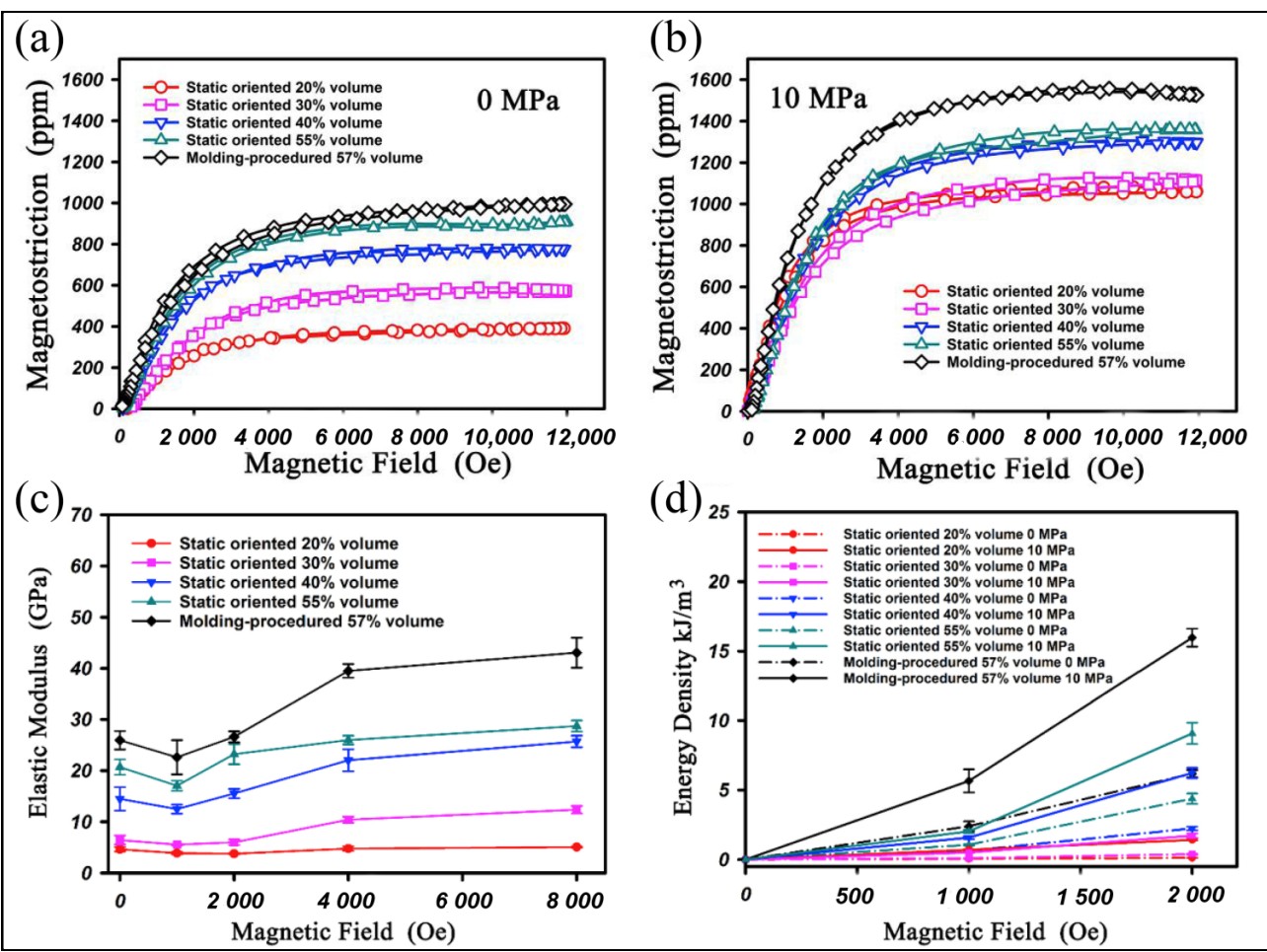

**Figure 22.** Magnetostriction curves of static magnetically oriented GMPCs and GMPCs subjected to a molding procedure under (**a**) 0 MPa and (**b**) 10 MPa uniaxial pressure; (**c**) elastic modulus curves and (**d**) energy densities of GMPC samples [87]. Reprinted with permission from ref. [87]. Copyright 2019 Elsevier.

### 6.2. Sintered Tb-Dy-Fe Material Composited with Dy-Cu Alloys

For Tb-Dy-Fe composites with polymers, a high orientation degree and a higher content of particles in the composites cannot be achieved simultaneously, which limits the further improvement of energy density.

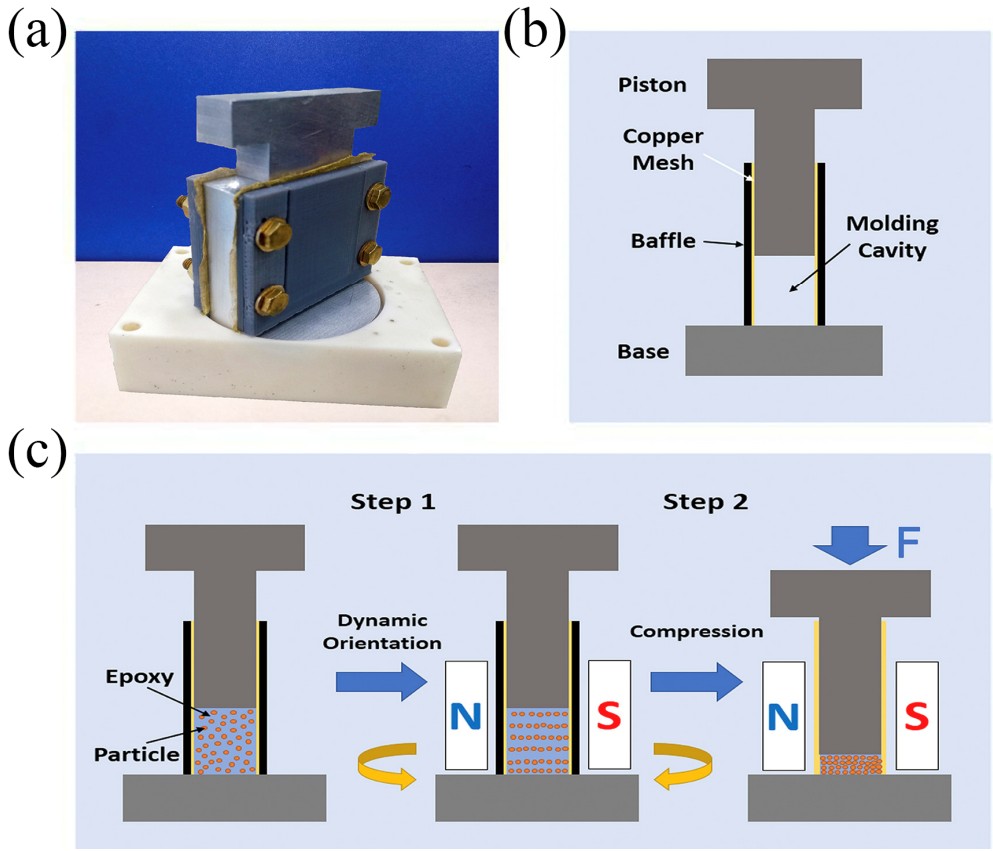

**Figure 23.** (**a**) The forming device; (**b**) schematic diagram of the forming device; (**c**) two-step molding procedure diagram [87]. Reprinted with permission from ref. [87]. Copyright 2019 Elsevier.

Recently, a new approach was proposed by Zhou et al. [90], which combined powder metallurgy with the magnetic field orientation. As shown in Figure 24, the powder slurry prepared with $Tb_{0.33}Dy_{0.67}Fe_{1.95}$ single-crystal particles and low-melting-point $DyCu_{1.6}$ alloy powders was wet pressed and oriented by a magnetic field. Subsequently, the compacts covered by $Dy_{1.2}Cu$ alloy thin ribbons were sintered at 1000 °C for 2 h. In this case, the $DyCu_{1.6}$ alloy powders acted as a "binder" to wet $Tb_{0.33}Dy_{0.67}Fe_{1.95}$ particles and provided a liquid channel for subsequent diffusion of $Dy_{1.2}Cu$ alloy, leading to an increase in the relative sintering density. Furthermore, a higher Tb-Dy-Fe particle content of above 90% was obtained in the sintered composites. Consequently, the <111> orientation and high sintering density effectively enhanced magnetostriction, as shown in Figure 25. More importantly, a major improvement of mechanical properties, with 176 MPa in bending strength and 71.3 MPa in tensile strength, was realized. This was attributed to the ductile Dy-Cu intergranular phase distributed along grain boundaries and the semicoherent interface between the Dy-Cu grain boundary phase and the brittle $(Tb,Dy)Fe_2$ matrix phase. The low-melting-point Dy-Cu phase was introduced as the new grain boundary phase to the Tb-Dy-Fe alloys using the sintering method, and the <111> orientation degree was improved by magnetic field orientation combined with the adjustment of the Tb/Dy ratio and particle morphology. As a result, high mechanical properties and high magnetostrictive properties were obtained in the sintered Tb-Dy-Fe/Dy-Cu composites. Mechanically, the bending strength, fracture toughness and tensile strength were respectively 3.67, 2.41 and 2.55 times those of the directionally solidified polycrystalline Tb-Dy-Fe alloy [90].

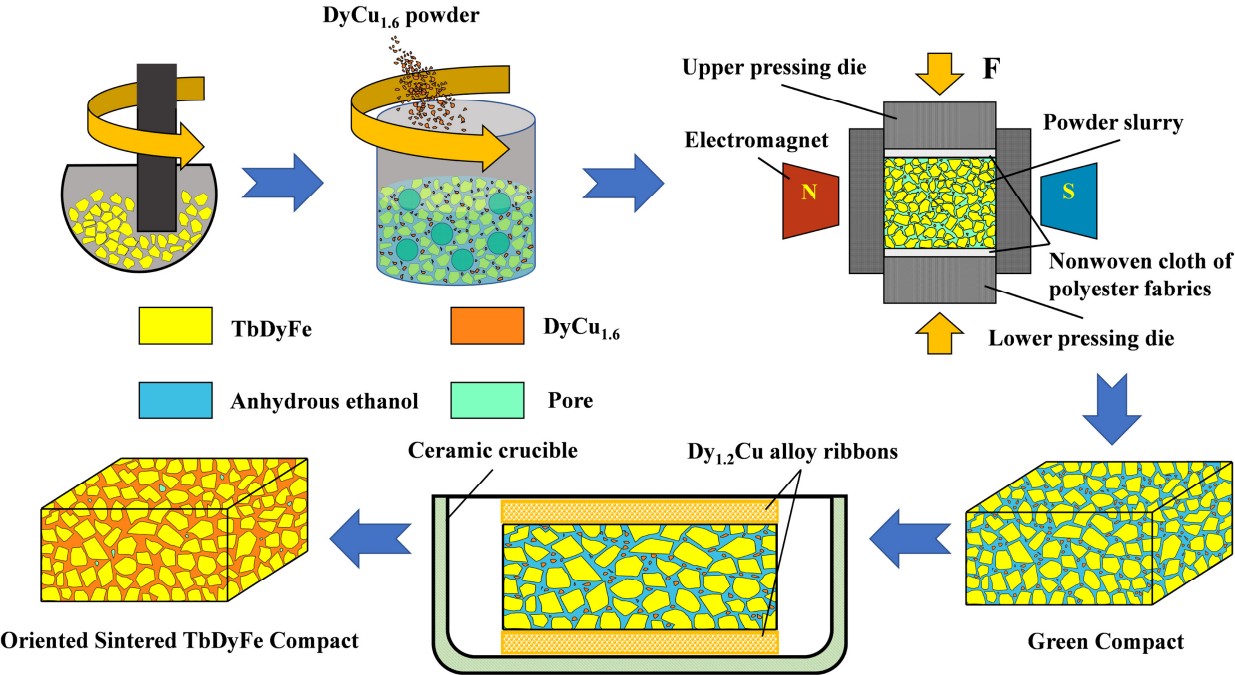

**Figure 24.** Schematic diagram of preparation procedure of the oriented sintered $Tb_{0.33}Dy_{0.67}Fe_{1.95}$ compacts [90]. Reprinted with permission from ref. [90]. Copyright 2022 Elsevier.

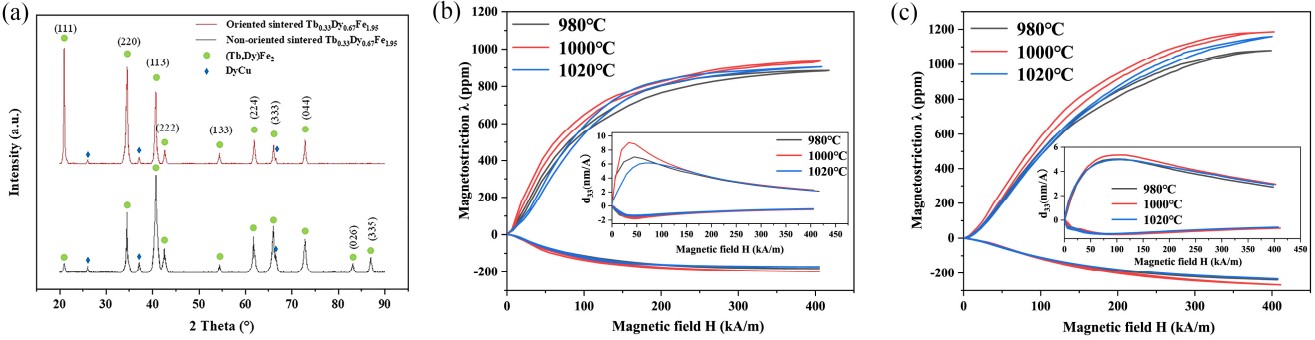

**Figure 25.** (**a**) XRD of the diffraction patterns of oriented and nonoriented sintered $Tb_{0.33}Dy_{0.67}Fe_{1.95}$ compacts at 1000 °C and $\lambda_{//}$, $\lambda_{\perp}$ magnetostriction curves, dependence of strain coefficient ($d_{33}$) of the oriented sintered $Tb_{0.33}Dy_{0.67}Fe_{1.95}$ compacts on magnetic field (H) under uniaxial pressure (**b**) 0 MPa, and (**c**) 10 MPa [90]. Reprinted with permission from ref. [90]. Copyright 2022 Elsevier.

## 7. Progress in Application of Tb-Dy-Fe Alloys

### 7.1. Tb-Dy-Fe Giant Magnetostrictive Thin Film

The research of small-scale magnetic structures is closely related to the design of microdevices and has attracted extensive attention. Tb-Dy-Fe thin film has certain applications in microelectromechanical systems (MEMS), such as microactuators and force sensors, because of its high sensitivity and large strain.

It was reported that films grown at higher substrate temperature have the combination of out-of-plane magnetic anisotropy and in-plane magnetic anisotropy [91]. Panduranga et al. obtained high-quality magnetoelastic film using sputtered Terfenol-D film after substrate heating and annealing crystallization at 450 °C [92]. In addition, another key problem of small-scale Tb-Dy-Fe materials is oxidation. It was found that the composition ratio of rare-earth oxides can be determined by the anomalous X-ray scattering method [93].

The magnetic properties of the films obtained by magnetron sputtering were similar to those of bulk single-crystal materials [92]. The magnetic properties of the micropatterned films obtained by photolithography and argon etching had little change relative to the

continuous films, which was considered to be related to the oxidative passivation of the sidewall, as well as the strong dipole pair ratio of 3μm MFM and the strong blue color in the 20μm PEEM image in Figure 26. The results showed that it has a pseudo-single domain structure [94].

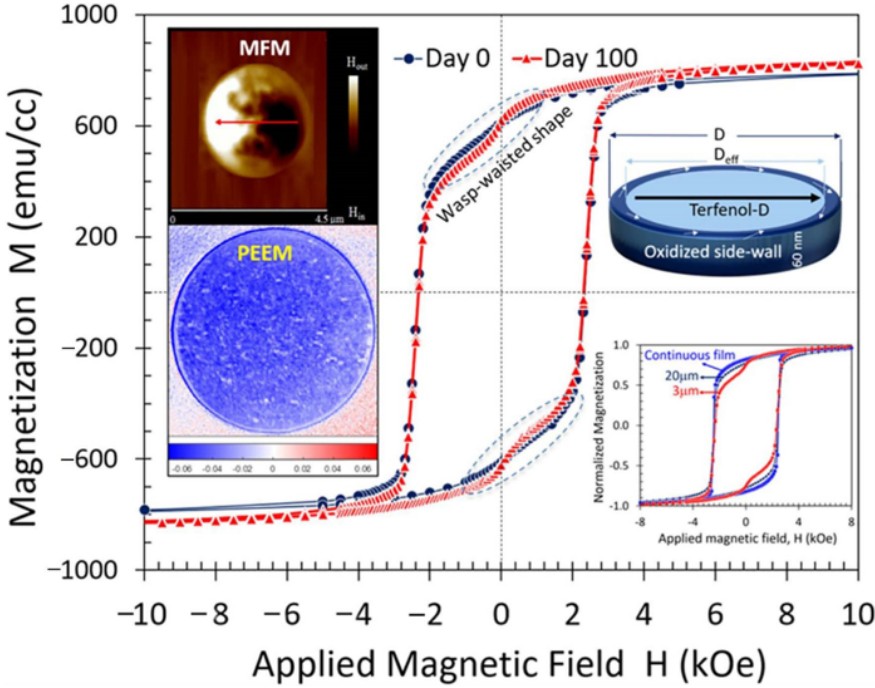

**Figure 26.** M vs. H plots of patterned Terfenol-D microdisks measured over the course of 100 days. Bottom-right inset shows the normalized M vs. H plots of continuous, 20 μm diameter disks and 3 μm diameter disks. Top-right inset shows the schematic of Terfenol-D disk with sidewall oxidation. Top-left inset shows magnetic force microscope image of an individual 3 μm disk upon initializing with an in-plane saturation magnetic field of 0.5 T and PEEM image of 20 μm disk indicating the single domain state when measured with Dy-$M_5$ edge [94]. Reprinted with permission from ref. [94]. Copyright 2021 Elsevier.

It was reported that the bimaterial cantilever structure of the magnetostrictive film prepared by electrodeposition had a large magnetostrictive coefficient of about 1250 ppm under an 11 kOe magnetic field (Figure 27), and the energy density was able to reach 100–165 kJ/m$^3$ [95]. In microdevices with a cantilever structure, a multilayer structure was generally used for better device performance. For instance, Tb-Dy-Fe/graphene/Tb-Dy-Fe multilayer film was used to replace the traditional three-layer Tb-Dy-Fe film in recent research, which resulted in a reduction in the dynamic response delay time.

### 7.2. Application in Microsensors and Other Devices

Magnetostrictive materials are usually used in actuators and sonar transducers. Recently, Tb-Dy-Fe materials were used in various high-sensitivity magnetostrictive sensors, such as current sensors [96,97], magnetic sensors [98] and torque sensors [99]. To explore its application, some researchers sputtered Tb-Dy-Fe film on a Fe-Co substrate, and the composite film was expected to be used in high-precision nondestructive testing [100]. In recent years, an application in wireless temperature measurement was developed, in which Terfenol-D was used to increase the temperature coefficient of resonant frequency [101].

A Tb-Dy-Fe magnetostrictive transducer combined with an optical fiber sensor is able to improve the accuracy of magnetic field detection. A series of high-precision magnetic field sensors were prepared by combining them with a fiber Bragg grating (FBG) sensor [102–104] and phase-shifted fiber Bragg grating (PS-FBG) [105]. The high-sensitivity response of the device was improved by the design of different systems. Feng et al. prepared

a high-finesse fiber-optic extrinsic Fabry–Perot interferometer (EFPI)-based sensor, and its sensitivity was significantly improved based on the design of a mechanical amplification structure [106].

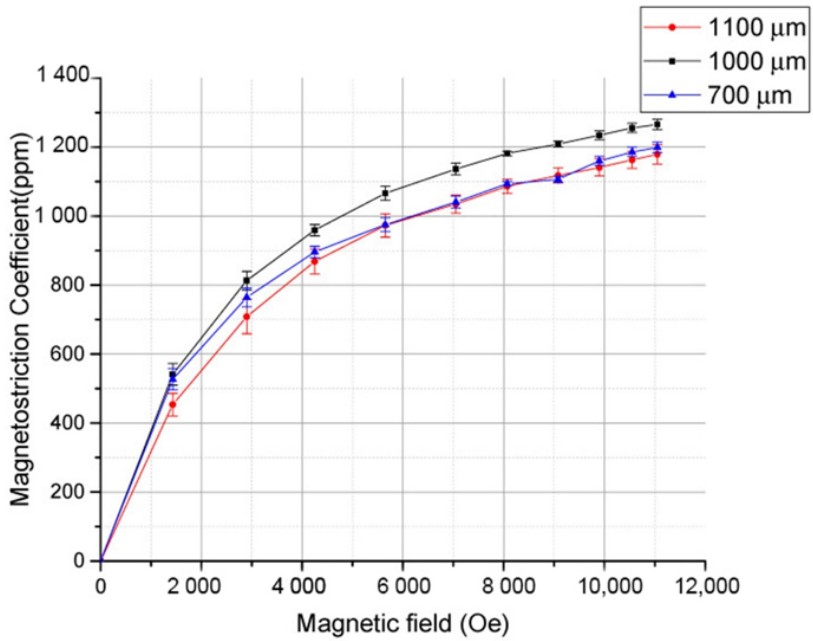

**Figure 27.** Magnetostriction coefficients of the $Tb_{0.36}Dy_{0.64}Fe_{1.9}$ film obtained from three cantilevers with lengths 700, 1000 and 1100 μm [95]. Reprinted from ref. [94].

## 8. Summary and Prospects

Improvements in the directional solidification preparation process, such as directional solidification with a strong magnetic field, are effective ways to obtain better orientation for directionally solidified Tb-Dy-Fe alloys. Tb-Dy-Fe composites prepared by the resin bonding method or low-melting-point alloy sintering method are more based on the improvement of high-frequency application properties and mechanical properties. At present, through the improvement of composite methods such as dynamic magnetic field orientation, the volume fraction and properties of Tb-Dy-Fe alloy particles have been greatly improved. In order to achieve better practical value, we need to constantly explore Tb-Dy-Fe series alloys with large magnetostriction, high mechanical properties, low loss and low cost. It is worth mentioning that the Dy-Cu phase was introduced as a new grain boundary phase to Tb-Dy-Fe alloys by the sintering method, and the <111> orientation was improved by the magnetic field orientation combined with the adjustment of the Tb/Dy ratio and particle morphology. High mechanical and magnetostrictive properties were achieved in sintered Tb-Dy-Fe/Dy-Cu composites. Meanwhile, sintered composite materials based on the reconstruction of the grain boundary phase also provide new ideas for the development of Tb-Dy-Fe materials with excellent comprehensive properties, including high magnetostriction, high mechanical properties, high corrosion resistance and high resistivity.

The studies of phase structure and ferromagnetic MPB, combined with domain structure, phase-field simulation and first-principles calculations, are very helpful to deeply understand the magnetostriction origin of Tb-Dy-Fe alloys. At the same time, such studies also facilitate an understanding of the influence of element substitution or developing a new alloy system. However, the mechanism and theoretical system still need more in-depth research to provide further support.

The research progress on the application of Tb-Dy-Fe alloys is extensive, especially in microdevices and various high-sensitivity sensors based on Tb-Dy-Fe films. However, bulk Tb-Dy-Fe materials could play a crucial role in a broader field on the basis of the effective improvement of comprehensive properties.

**Author Contributions:** Conceptualization, Z.Y. and J.L.; methodology, Z.Y. and J.L.; validation, Z.Y. and J.L.; formal analysis, Z.Y. and J.L.; investigation, Z.Y. and J.L.; resources, Z.Y. and J.L.; data curation, Z.Y. and J.L.; writing—original draft preparation, Z.Y. and J.L.; writing—review and editing, Z.Y., J.L., Z.Z. and J.G.; visualization, Z.Y. and J.L.; supervision, J.L. and X.G.; project administration, X.G.; funding acquisition, X.B. and X.G. All authors have read and agreed to the published version of the manuscript.

**Funding:** This research was funded by the National Key R & D Program of China, grant number 2021YFB3501403; the State Key Laboratory for Advanced Metals and Materials, grant numbers 2017Z-11 and 2018Z-07; and Fundamental Research Funds for the Central Universities, grant numbers FRF-GF-17-B2, FRF-GF-19-028B and FRF-GF-20-23B.

**Institutional Review Board Statement:** Not applicable.

**Informed Consent Statement:** Not applicable.

**Data Availability Statement:** Data available in a publicly accessible repository.

**Acknowledgments:** This work is supported by the National Key R & D Program of China (grant no. 2021YFB3501403), the State Key Laboratory for Advanced Metals and Materials (2017Z-11, 2018Z-07), and the Fundamental Research Funds for the Central Universities (FRF-GF-17-B2, FRF-GF-19-028B, and FRF-GF-20-23B).

**Conflicts of Interest:** The authors declare no conflict of interest.

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
