# Peer review of "Recent Advances in Magnetostrictive Tb-Dy-Fe Alloys"

_metals, doi:10.3390/met12020341_

Round 1

Reviewer 1 Report

The paper is interesting a deserves to be published. Research design is complete and well described. But... English should be corrected. Long sentences make the text difficult to read.

Author Response

We would like to thank for the constructive comment, and have revised manuscript carefully. Many long sentences have been carefully modified. Any revisions made to the manuscript have been marked up using the “Track Changes” function of MS Word.

Reviewer 2 Report

Manuscript #metals-1559370

Title: Recent advances in magnetostrictive Tb-Dy-Fe alloys

Authors: Zijing Yang, Jiheng Li, Zhiguang Zhou, Jiaxin Gong, Xiaoqian Bao  and Xuexu Gao  

This paper contains enough significant new physics, and scientifically sounds to warrant publication in Metals.

The topic is very interesting, and the manuscript is generally well written, easy to follow and well organized.

The main conclusion aligns well with the experimental results.

Author Response

We would like to thank for the comment. The manuscript has been carefully revised. Any revisions made to the manuscript have been marked up using the “Track Changes” function of MS Word.

Reviewer 3 Report

Manuscript ID: metals-1559370

Title: Recent advances in magnetostrictive Tb-Dy-Fe alloys

The manuscript is well written and organized. Timely and relates on a topic which is of great interest.

In my opinion the paper could be published as it stands.

Author Response

(The authors gave the same response as above.)

Reviewer 4 Report

no comment, useful review article

Author Response

(The authors gave the same response as above.)
